J Physiol 603.11 (2025) pp 3445–3461

# Non-homogeneous distribution of inhibitory inputs among motor units in response to nociceptive stimulation at moderate contraction intensity

François Hug[1,2] (ID), François Dernoncourt[1], Simon Avrillon[1] (ID), Jacob Thorstensen[2,3] (ID), Manuela Besomi[4], Wolbert van den Hoorn[4,5] (ID) and Kylie Tucker[2] (ID)

[1] Université Côte d'Azur, LAMHESS, Nice, France
[2] School of Biomedical Sciences, The University of Queensland, Brisbane, QLD, Australia
[3] Faculty of Health Sciences & Medicine, Bond University, Gold Coast, Australia
[4] Centre of Clinical Research Excellence in Spinal Pain, Injury and Health, School of Health and Rehabilitation Sciences, The University of Queensland, Brisbane, QLD, Australia
[5] School of Exercise and Nutrition Sciences, Queensland University of Technology, Brisbane, QLD, Australia

Handling Editors: Richard Carson & Madeleine Lowery

The peer review history is available in the Supporting information section of this article (https://doi.org/10.1113/JP288504#support-information-section).

**Abstract figure legend** We combined experimental data and *in silico* models to investigate the contribution of inhibitory and neuromodulatory inputs to motor unit behaviour in response to nociceptive stimulation during submaximal isometric contractions at 30% of maximal voluntary contraction. We identified large samples of motor units in the vastus lateralis, leading to three key observations. First, while motor unit discharge rates significantly decreased during Pain, a substantial proportion of motor units did not show this decrease and, in some cases, even exhibited an increase. Second, using complementary approaches, we found that pain did not significantly affect neuromodulation, making it unlikely to be a major contributor to the observed changes in motor unit behaviour. Third, we observed a significant reduction in the proportion of common inputs to motor units during Pain. Together with our simulations, these results provide evidence of increased inhibition that is non-uniformly distributed across motor units, regardless of their size. ppp, pulses per second; MVC, maximal voluntary contraction.

This article was first published as a preprint. Hug F, Dernoncourt F, Avrillon S, Thorstensen J, Besomi M, van den Hoorn W, Tucker K. Non-Homogeneous Distribution of Inhibitory Inputs Among Motor Units in Response to Nociceptive Stimulation. bioRxiv. https://doi.org/10.1101/2024.10.05.616762

**Abstract**  Pain significantly influences movement, yet the neural mechanisms underlying the range of observed motor adaptations remain unclear. This study combined experimental data and *in silico* models to investigate the contribution of inhibitory and neuromodulatory inputs to motor unit behaviour in response to nociceptive stimulation during contractions at 30% of maximal torque. Specifically, we aimed to unravel the distribution pattern of inhibitory inputs to the motor unit pool. Seventeen participants performed isometric knee extension tasks under three conditions: Control, Pain (induced by injecting hypertonic saline into the infra-patellar fat pad) and Washout. We identified large samples of motor units in the vastus lateralis (up to 53/participant) from high-density electromyographic signals, leading to three key observations. First, while motor unit discharge rates significantly decreased during Pain, a substantial proportion of motor units (14.8–24.8%) did not show this decrease and, in some cases, even exhibited an increase. Second, using complementary approaches, we found that pain did not significantly affect neuromodulation, making it unlikely to be a major contributor to the observed changes in motor unit behaviour. Third, we observed a significant reduction in the proportion of common inputs to motor units during Pain. To explore potential neurophysiological mechanisms underlying these results, we simulated the behaviour of motor unit pools with varying distribution patterns of inhibitory inputs. Our simulations support the hypothesis that a non-homogeneous distribution of inhibitory inputs, not strictly organised according to motor unit size, is a key mechanism underlying the motor response to nociceptive stimulation during moderate contraction intensity.

(Received 13 January 2025; accepted after revision 12 May 2025; first published online 31 May 2025)

**Corresponding author** F. Hug: Université Côte d'Azur, LAMHESS, Nice, France.    Email: Francois.hug@univ-cotedazur.fr

## Key points

- Pain affects movement, but the neural mechanisms underlying these motor adaptations are not well defined.
- The traditional view is that pain causes uniform (homogeneous) inhibition among motor units.
- Recent research has observed differential motor unit responses to experimental pain – some with decreased discharge rates and others with increased discharge rates.
- Combining experimental data with modelling, we provide compelling evidence of increased inhibition that is non-uniformly distributed across motor units, regardless of their size.

## Introduction

Pain affects movement, but the neural mechanisms underlying these motor adaptations are not well defined. As the final common pathway for movement control (Sherrington, 1906), spinal motor neurons transmit neural signals from various sources to muscle fibres.

As such, measuring spinal motor neuron activity during nociceptive stimulation offers valuable insights into the motor adaptations to pain. In humans, this has been achieved by recording the spiking activity of single motor units (comprising a spinal motor neuron and the muscle fibres it innervates) in the presence of experimentally induced pain. Most studies report an overall reduction

**François Hug** received his PhD in Human Movement Sciences from Aix-Marseille Université, France (2003). He was a Full Professor at Nantes Université before joining Université Côte d'Azur in 2021, where he currently leads the LAMHESS lab. He is an honorary fellow of the Institut Universitaire de France and an honorary professor at The University of Queensland, Australia. His research focuses on the neural control of movement, with recent work emphasizing motor unit population-level analysis. He has (co-)authored over 200 peer-reviewed publications. Since 2022, he has served on the council of the International Society of Electrophysiology and Kinesiology (ISEK).

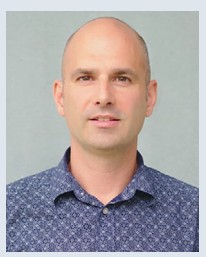

in motor unit discharge rates in muscles that are closely associated with the painful region (Farina et al., 2005; Sohn et al., 2000; Tucker & Hodges, 2009; Tucker et al., 2009). However, this overall (average) decrease in motor unit discharge rate, observed during low- to moderate-intensity force-matched contractions, is often accompanied by an increase in the discharge rate of a small proportion of motor units and the recruitment of new units (Hug et al., 2013; Martinez-Valdes et al., 2021; Tucker et al., 2009). This multifaceted motor response to pain is thought to assist with the maintenance of force output; however, the neural basis for these differential effects of nociceptive stimulation across motor units remains unknown.

The current literature does not clearly establish whether the differential effects of nociceptive stimulation arise from homogeneous or non-homogeneous patterns of inhibition. Martinez-Valdes et al. (2020) observed a divergent response to nociceptive stimulation between low- and high-threshold motor units during contractions performed at a relatively high intensity (70% of maximal voluntary contraction (MVC)). They proposed that this divergent effect arises from a non-homogeneous distribution of inhibitory afferent inputs, with greater inhibition directed toward low-threshold units. It is important to note that these results are also compatible with a homogeneous distribution of inhibitory inputs, as such homogeneous inhibition would cause a greater inhibitory post-synaptic potential in low-threshold units (Luscher et al., 1979). Either way, to compensate for inhibition and maintain force, increased excitatory input to the motoneuron pool may occur (Hodges & Tucker, 2011), potentially explaining the increased discharge rate of higher-threshold units (Martinez-Valdes et al., 2020). Notably, the divergent responses between low- and high-threshold units could also involve differential changes in motor neuron intrinsic properties (Mesquita et al., 2020). For example, the reticular activating system, involved in the transmission and modulation of nociceptive information, releases monoamines (Martins & Tavares, 2017), which may stimulate persistent inward currents. These currents amplify and extend the effects of synaptic input, and appear to have a greater influence on the discharge of low- than high-threshold units (Avrillon et al., 2023; Lee & Heckman, 1998). As persistent inward currents are sensitive to inhibition (Hultborn et al., 2003; Hyngstrom et al., 2007; Revill & Fuglevand, 2017), homogeneous inhibition to the motor unit pool could have a greater impact on the persistent inward currents – and thus the discharge rate – of low-threshold units (Mesquita et al., 2020). In addition, there are observations that newly recruited higher-threshold units during pain do not always follow the expected orderly recruitment pattern (Tucker et al., 2009). Combined with evidence of differential changes in motor neuron excitability within

the pool (Hodges et al., 2021), these deviations from the typical orderly recruitment pattern could suggest a non-homogeneous inhibition that is not strictly organised according to motor unit size.

Unravelling the distribution pattern of inhibition across motor units requires the activity of a large sample of motor units to be recorded, which is not feasible using traditional methods that use a limited number of intramuscular electrodes (Hug et al., 2013; Sohn et al., 2000; Tucker et al., 2009). In this study, we identified a large sample of motor units (up to 53 per participant per contraction) by decomposing high-density EMG signals collected during moderate-intensity isometric contractions. By combining experimental data and *in silico* models, we aimed to quantify the contribution of inhibitory and neuromodulatory inputs to the behaviour of motor unit populations during experimental joint pain.

## Methods

### Participants and ethical approval

Seventeen volunteers (two females; mean $\pm$ standard deviation; age: $27.0 \pm 6.5$ years, height: $176.0 \pm 9.1$ cm, and body mass: $73.5 \pm 14.3$ kg) participated in this study. They had no history of lower leg pain that limited function, required time off work or physical activity, or necessitated consultation with a health practitioner in the 6 months prior to testing. The procedures were approved by the local ethics committee (The University of Queensland; 2021/HE002253), and conducted in accordance with the *Declaration of Helsinki*, except for registration in a database. The participants were fully informed of any risks or discomfort associated with the procedures before providing written informed consent to participate. Notably, no motor units were identified in two male participants. Consequently, the results are reported for the remaining 15 participants.

### Experimental design

Participants sat on a supportive chair, with their hips flexed at 80° (0° being the neutral position) and the right knee flexed at 80° (0° being the full extension) (Fig. 1). Isometric force was measured using a three-dimensional force gauge (MC3A, AMTI, USA). Inextensible straps were tightened to immobilize their torso and pelvis.

The experimental session began with a standardized warm up. After 2 min of rest, the participants performed two maximal voluntary contractions for 3 s, with 60 s of rest in between. A third maximal contraction was performed if the difference in force between the first two contractions was greater than 10%. Peak MVC force was

considered as the maximal value obtained from a moving average window of 250 ms.

The rest of the experimental session consisted of a series of isometric submaximal contractions under three different conditions: Control, during the presence of experimental pain (Pain) and after pain had ceased (Washout). For each condition, participants performed one triangular force contraction (10 s ramp up and 10 s ramp down) to 30% MVC and three trapezoidal force contractions (6 s ramp up followed by 20 s plateau at 30% MVC) interspaced by 30 s of rest (Fig. 1). Using real-time force feedback displayed on a monitor, the participants were asked to match the target force. Because estimation of the magnitude of persistent inward currents requires that the participants accurately match the target force, an examiner visually assessed the match between the produced force and the triangular target force in real time. If the produced force showed noticeable deviations from the expected triangular shape, such as irregular slopes or excessive fluctuations, an additional triangular contraction was performed. To limit this risk, participants were first familiarized with the tasks.

To minimize the impact of the preceding contractions on persistent inward currents, a 2 min rest period was systematically provided before the triangular contractions. At the beginning of this rest period, participants briefly contracted the antagonist muscles (knee flexion), as such contraction effectively attenuates persistent inward currents (Gomes et al., 2024) via reciprocal inhibition. Because pain induced by hypertonic saline injection typically lasts less than 5 min (Martinez-Valdes et al., 2020), the triangular contraction was always performed first during the Pain condition. This sequence avoided the additional 2 min rest period required if trapezoidal contractions were performed first, thus ensuring that all contractions were performed while significant (greater than 2/10) pain levels were experienced.

## Surface electromyographic recordings

To record a large sample of motor units, we selected the vastus lateralis, which allowed the placement of four adhesive grids of 64 electrodes each (total of 256 electrodes; inter-electrode distance: 8 mm; GR08MM1305, OT Bioelettronica, Italy). The grids were positioned over the

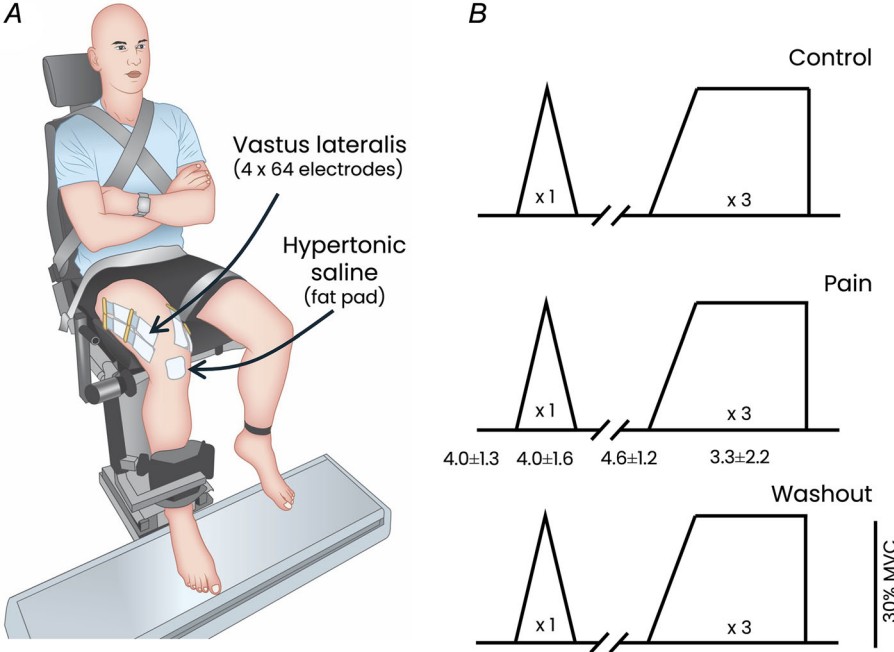

**Figure 1. Experimental setup and protocol**
*A*, participants sat on a supportive chair, with their hips flexed at 80° (0° being the neutral position) and the right knee flexed at 80° (0° being the full extension). Isometric force was measured using a three-dimensional force gauge. *B*, the experimental tasks consisted of a series of isometric submaximal contractions under three different conditions: Control, during the presence of experimental pain (Pain) and after pain had ceased (Washout). For each condition, participants performed one triangular force contraction (10 s ramp up and 10 s ramp down) to 30% maximal voluntary contraction (MVC) and three trapezoidal force contractions (6 s ramp up followed by 20 s plateau at 30% MVC) interspaced by 30 s of rest. Four adhesive grids of 64 electrodes (total of 256 electrodes) were placed over the vastus lateralis muscle. Pain was induced by a single bolus injection of hypertonic saline (0.35 ml, 5.8% NaCl) into the infra-patellar fat pad. The mean pain intensity is shown on the middle (pain) panel. Panel *A* adapted from Avrillon et al. (2021). [Colour figure can be viewed at wileyonlinelibrary.com]

muscle belly to cover the largest surface while avoiding the boundaries identified by manual palpation. This setup was chosen to maximize the motor unit yield (Avrillon et al., 2023; Caillet et al., 2023).

Before electrode application, the skin was shaved and cleaned with an abrasive gel (Nuprep, Weaver and company, USA). The adhesive grids were held on the skin using bi-adhesive foam layers (OT Bioelettronica, Italy). Skin–electrode contact was ensured by filling the cavities of the adhesive layers with a conductive paste (AC cream, Spes Medica, Genova, Italy). A 10 cm wide elastic band was placed over the electrodes with a slight tension to ensure that all the electrodes remained in contact with the skin throughout the experiment. A strap electrode dampened with water was placed around the contra-lateral ankle (ground electrode) and a reference electrode (5 × 5 cm, Kendall Medi-Trace, Canada) was positioned over the patella of the tested limb. The EMG signals were recorded in monopolar mode, bandpass-filtered (10–500 Hz) and digitized with a 16-bit precision at 2048 samples/s using a multichannel acquisition system (EMG-Quattrocento; 400-channel EMG amplifier, OT Bioelettronica, Italy).

### Experimental pain

Pain was induced by a single bolus injection of hypertonic saline (0.35 ml, 5.8% NaCl) into the infra-patellar fat pad (Tucker & Hodges, 2009) using a 25G × 16 mm needle. This injection site was chosen to avoid any direct impact of hypertonic saline on axons other than small-diameter pain fibres, including the motor neurons themselves. Participants rated pain intensity throughout the Pain condition on an 11-point numerical rating scale, anchored with 'no pain' at 0 and 'maximum imaginable pain' at 10. After the Pain condition, participants drew the area of pain on a picture of their knee.

### Data analysis

**Amplitude of the interference EMG signals.** For each of the four grids, the 64 EMG signals were visually inspected for noise and artefacts. Signals identified as noisy or containing artefacts were excluded from further analysis and were replaced by the linear interpolation of all the adjacent channels (between two and four depending on the location of the electrode). Then, single differential signals were calculated as the differences between adjacent electrodes along the column direction, resulting in 59 signals per grid. These signals were rectified and averaged to provide a mean EMG amplitude value.

**Decomposition of EMG signals.** We used the convolutive blind source separation method (Negro, Muceli et al.,

2016) implemented in an open-source software (MUedit (Avrillon et al., 2024)) to decompose the EMG signals into motor unit spiking activity. The automatic decomposition was performed on the monopolar EMG signals (noisy channels excluded) from one contraction per task and condition. Then, we manually edited all the motor unit spike trains following previously published procedures (Del Vecchio et al., 2020; Hug et al., 2021). Importantly, this procedure has demonstrated high reliability across operators (Hug et al., 2021) and has been validated against simulated signals (Rossato et al., 2024).

Since each trapezoidal force-matching task included three contractions, we applied the edited motor unit filters from the decomposed contraction to the entire signal. We then manually edited all motor units in each trapezoidal contraction. The editing process involved an iterative approach, discarding detected peaks that result in erroneous discharge rates (outliers) and adding missed discharge times that are clearly distinguishable from the noise. The motor unit pulse trains were recalculated with updated motor unit filters and were accepted by the operator once all the putative discharge times were selected. For more details on the manual editing step, please see Avrillon et al. (2024).

**Motor unit tracking.** EMG decomposition was performed independently on each of the four electrode grids placed over the vastus lateralis muscle. As such, the same motor units could theoretically be detected across multiple grids. To identify duplicate units, we compared their spike trains. Specifically, the spike trains of each pair of motor units were initially aligned using a cross-correlation function to account for any potential delay caused by the propagation of action potentials along the fibres. Then, discharge times that occurred within 0.5 ms (one sample) intervals were considered common between motor units, and motor units sharing more than 30% of their discharge times were considered to be duplicates. In cases where duplicate units were found, only the motor unit with the lowest coefficient of variation of its inter-spike intervals was retained.

We also tracked motor units across the three conditions (Control, Pain and Washout) for each task (trapezoidal and triangular contractions) separately. To achieve this, the motor unit filters identified in one condition were applied to the extended and whitened EMG signals of the two other conditions (Francic & Holobar, 2021). For each of these two conditions, the resulting spike trains were compared with those initially obtained. A motor unit was considered tracked if at least 30% of the discharge times were shared. This process was applied to every potential pair of conditions. Only the motor units tracked across the three conditions were included in the analyses.

**Motor unit discharge characteristics.** For each motor unit, the recruitment threshold was expressed as a percentage of MVC force. For the triangular contractions, recruitment time was defined as the time of the first firing. For the trapezoidal contractions, we considered the average recruitment threshold across the three contractions. We also calculated the mean discharge rate over the three torque plateaus of the trapezoidal contractions. Finally, the cumulative spike train was computed by summing the spike trains of all active motor units. It was then low-pass-filtered at 2.5 Hz and correlated with the effective force.

**Estimates of persistent inward currents.** First, we estimated the magnitude of persistent inward currents using a paired motor unit analysis applied to the triangular contractions (Gorassini et al., 2002). To achieve this, we smoothed the instantaneous discharge rate of each motor unit using a support vector regression (Beauchamp et al., 2023). Then, we estimated the discharge rate of a lower-threshold motor unit (control unit) at the time of recruitment and de-recruitment of a higher-threshold motor unit (test unit). In this approach, the discharge rate of the control unit is used as a proxy for the net synaptic input to the test unit (Gorassini et al., 1998, 2002). The difference in instantaneous discharge rate between the recruitment and de-recruitment times of the test unit, referred to as $\Delta F$, is considered proportional to the magnitude of persistent inward currents (Gorassini et al., 2002). $\Delta F$ values were considered only for pairs of motor units that met the following criteria (Hassan et al., 2020): (i) the test motor unit must discharge for at least 2 s, (ii) the test motor unit must be recruited at least 1 s after the control motor unit to ensure full activation of persistent inward currents, (iii) the test motor unit must be de-recruited at least 1.5 s prior to the control motor unit to prevent overestimation of $\Delta F$, and (iv) the smoothed discharge rate of the test and control motor units must be correlated, with a coefficient of determination ($R^2$) $\geq$ 0.7. Each reported $\Delta F$ value represents the average value across all pairs involving a given test motor unit with all possible control units (on average: 5 $\pm$ 3 combinations; range: 1–15). As we tracked motor units across conditions, the reported $\Delta F$ values were derived from the same pairs of motor units across the three conditions (Control, Pain and Washout).

Because $\Delta F$ values depend on both neuromodulation and inhibition supplied to the pool of motor units (Beauchamp et al., 2023; Revill & Fuglevand, 2017), we used complementary metrics to provide a more direct estimate of the neuromodulatory inputs. Specifically, for each motor unit, we quantified the deviation from a linear increase in discharge rate relative to actual torque during the ramp-up phase of the triangular contra-

ctions (Beauchamp et al., 2023). First, we fitted a straight line from the discharge rate at recruitment to the peak discharge rate. We calculated the brace height as the magnitude of the longest vector orthogonal to this straight line, extending to the smoothed discharge rate. To account for different recruitment thresholds across motor units, and thus different ranges of discharge rates, the brace height value was normalized to the height of a right triangle whose hypotenuse represents a straight line from recruitment to peak discharge (Beauchamp et al., 2023). We also quantified the acceleration as the slope of the linear increase between the first discharge and the instance at which the brace height occurred, and the attenuation as the slope of the linear increase between the instance at which the brace height occurred and the maximal discharge rate. Acceleration corresponds to an amplification of synaptic inputs, likely due to the activation of persistent inward currents (Lee & Heckman, 2000).

**Estimation of common synaptic inputs.** We used two different approaches to estimate the level of dominant common synaptic inputs received by the pool of vastus lateralis motor units. These analyses were conducted on the concatenated plateaus of the trapezoidal contractions, including only the motor units that fired continuously (i.e. no pause in firing longer than 250 ms).

First, we estimated the proportion of common input with respect to the total input received by the motor neuron pool (Negro, Yavuz et al., 2016). Specifically, we calculated the coherence between two equally sized groups of cumulative spike trains using the Neurospec toolbox (Halliday, 2015). The number of motor units in each of the two groups varied from one to the maximum number (half of the total number of identified units) and 100 random permutations of the identified units were performed for each iteration. We modelled the relationship between the average coherence values in the 0–5 Hz bandwidth and the number of motor units in each group using a least-squares curve fitting approach, based on the two-parameter model described by Negro, Yavuz et al. (2016). One parameter reflects the contribution of the common synaptic input, while the other reflects the total synaptic input. The square root of the ratio between these two parameters provides an estimate of the proportion of common synaptic input.

Second, we applied a principal component analysis to the smoothed discharge rate of motor units. All binary motor unit spike trains were convoluted with a 400 ms Hanning window, such that only the oscillations of the signal related to the fluctuation of force were considered (Negro et al., 2009). These signals were high pass, bi-directionally filtered to remove offsets and trends using a Butterworth filter second order with a cut-off frequency of 0.75 Hz. We normalized the smoothed discharge rates

between 0 and 1 to avoid any bias to the motor units with a higher greater change in discharge rate. Finally, normalized smoothed discharge rates were concatenated in a $n \times T$ matrix, where $n$ is the number of motor units and $T$ the number of time samples. We applied principal component analysis to this matrix and quantified the variance explained by the first principal component.

### *In silico* model

To explore potential neurophysiological mechanisms underlying the experimental results, we simulated the behaviour of motor unit pools with different distributions of inhibitory inputs, as detailed below. To this end, we used a spiking neural network model implemented in Python (Brian 2 (Stimberg et al., 2019)). The code and the list of parameters are publicly available at: https://github.com/FrancoisDernoncourt/Pain_inhibition_simulation

**Leaky integrate-and-fire equations.** We modelled a pool of 300 motor units using a conductance-based leaky integrate-and-fire model. Each motor neuron had a resting membrane potential and reset voltage $E_{rest}$ of 0 mV and a firing threshold $V_{thresh}$ of 10 mV. The change in voltage per timestep $\frac{dv}{dt}$ was calculated as a function of the leak current $I_{leak}$ (in nA), excitatory current $I_{excit}$ (in nA), inhibitory current $I_{inhib}$ (in nA), and the membrane capacitance $C$ (in F), as follows:

$$\frac{dv}{dt} = \frac{-(I_{leak} + I_{excit} + I_{inhib})}{C} \qquad (1)$$

The current values were calculated according to the specific conductance values ($g$) and equilibrium potential ($E$), as follows:

$$I_{leak} = g_{leak} \times (v - E_{rest}) \qquad (2)$$

$$I_{excit} = \max(g_{excit} \times (v - E_{excit}) + I_{rheobase}, 0) \qquad (3)$$

$$I_{inhib} = g_{inhib} \times (v - E_{inhib}) \qquad (4)$$

An offset $I_{rheobase}$ was applied to the excitatory current, which corresponds to the minimum current necessary for the excitatory conductance to be non-zero. $E_{excit}$ was set to 25 mV, and $E_{inhib}$ to −15 mV. These values were chosen so that excitatory and inhibitory inputs induced a voltage change of the same magnitude when the membrane potential was halfway between $E_{rest}$ and $V_{thresh}$.

The conductance values (in mS) were defined as:

$$g_{leak} = \frac{1}{R_i} \qquad (5)$$

$$g_{excit} = \gamma_{excit}(t, i) \times S_i \qquad (6)$$

$$g_{inhib} = \gamma_{inhib}(t, i) \times S_i \qquad (7)$$

where $R_i$ (in Ohms) represents the membrane resistance of a given motor neuron $i$, $\gamma_{excit}(t, i)$ and $\gamma_{inhib}(t, i)$ are the time-dependent excitatory and inhibitory synaptic inputs delivered to each motor neuron $i$ at each timestep $t$, and $S_i$ is a scaling factor representing the responsiveness of the membrane potential to a given input. $S_i$ values were calculated as the normalized resistance of motor neurons, with $S_i$ of the smallest motor neuron set to 1. After firing, motor neurons underwent a refractory period $T_{refractory}$ during which the membrane potential was clamped to 0 mV.

**Motor unit properties.** Each motor neuron $i$ within the pool was assigned a specific soma diameter $D_i$ (in μm), which was assigned according to the following quadratic function, ranging from 50 μm ($D_{min}$) to 100 μm ($D_{max}$):

$$D_i = D_{min} + \left(\frac{i}{N}\right)^2 \times (D_{max} - D_{min}) \qquad (8)$$

This distribution ensured a higher representation of low-threshold motor neurons (Duchateau & Enoka, 2022). The relationship between the soma diameter and $R$, $C$, $I_{rheobase}$ and $T_{refractory}$ was determined using the equations provided by Caillet et al. (2022) (for additional details, please see the code).

To simulate the torque profiles generated by the active motor units, each unit was associated with specific twitch properties, with the torque of a single twitch ranging from 5 mN m (smallest unit) to 140 mN m (largest unit) and a time-to-peak ranging from 20 ms (largest unit) to 80 ms (smallest unit) (Van Cutsem et al., 1997). Both twitch torque and time-to-peak values were determined from linear interpolation based on the normalized soma diameter. Note that the ratio between a single twitch torque and tetanus torque ranged from 0.2 (smallest motor unit) to 0.3 (largest motor unit) (Brown & Loeb, 2000). The total torque output of the motor unit pool was obtained by summing the absolute torque profiles of all active units. A low-pass filter (fifth order Butterworth, bi-directional, cut-off: 10 Hz) was applied to the total force output to account for the mechanical impedance of the muscle–tendon unit.

**Input to the simulated neurons.** The simulated motor units received common excitatory input, common noise (0–5 Hz bandwidth) and independent noise (0–50 Hz bandwidth). The independent and common noise were modelled as low-pass-filtered Gaussian noise with a mean of 0 mS and a standard deviation of 0.03 and 0.015 mS, respectively. All the motor units within the pool received the same common excitatory input and common noise, while each motor unit received different

independent noise. We estimated the common excitatory input signal required for active motor units to match a target force plateau at 30% of MVC. This process involved an optimization procedure with the Adam optimizer (Kingma & Ba, 2014).

We tested six simulation scenarios. First, motor units received no inhibitory input (Control condition). Second, all motor units received a common inhibitory signal with the same weight (Homogeneous condition). Third, a common inhibitory signal was distributed to the motor unit pool following a decaying exponential function, with inhibition weights biased toward the low-threshold units (Gradient inhibition condition). A small standard deviation of 0.1 was added to account for slight variability in the weights. This condition was designed to test the hypothesis proposed by Martinez-Valdes et al. (2020) of differential inhibitory input between low- and high-threshold motor units. Based on our results showing no systematic correlation between the change in discharge rate during pain and the recruitment threshold (see *Results* section), the other scenarios were designed to test a heterogeneous distribution of inhibitory inputs to the motor unit pool. Specifically, a common inhibitory signal was assigned to motor units with random weights following a Gaussian distribution with either low (SD = 0.2) or high (SD = 0.5) variance (Heterogeneous inhibition conditions). In the final scenario, motor units were randomly divided into two groups. A common inhibitory signal was distributed to these groups, with a weight of 1.5 assigned to the first group, comprising 75% of the motor units, and a weight of 0.5 assigned to the second group, comprising 25% of the motor units (Clustered inhibition conditions). This ensured that the majority of motor units received a higher level of inhibition. The motivation for simulating clusters of inhibition stems from recent studies suggesting that motor units from the same pool may be grouped into different synergies based on the distinct net inputs they receive (Hug et al., 2023).

The mean level of inhibition was maintained to be constant across the five scenarios with inhibition, and the force was kept constant. To achieve this, the optimization procedure to determine the excitatory input (as described above) was performed after applying inhibition. Also, to account for the biased excitatory inputs toward larger motor neurons (Heckman & Enoka, 2012), the weight of the excitatory inputs was slightly adjusted, with lower values assigned to smaller motor units and higher values to larger ones, and linearly interpolated between them.

While only one simulation was run for the Control condition, 10 simulations were run for the conditions with inhibition to account for slightly different distribution on inhibitory inputs across the simulations. Simulations were conducted over a 60 s period, corresponding to the three 20 s plateaus in the experimental setup. We pre-processed and analysed the simulated spike trains of the active motor units using the same methods as for the experimentally recorded spike trains (see above). Of note, only the motor units that fired continuously were included in the analysis.

## Statistics

All statistical analyses were performed with RStudio (USA). First, the normality of the data distribution was assessed using quantile–quantile plots and histograms. Separate linear mixed-effects models, implemented in the *lmerTest* package, were applied to compare the outcomes across conditions. In these models, 'condition' (Control, Pain, Washout) was treated as a fixed effect, and 'participants' were modelled as random intercepts. When necessary, multiple comparisons were performed using the *emmeans* package, using the Tukey method to adjust the *P*-values.

We ran Pearson correlations to examine the relationship between the change in motor unit discharge rate across conditions and either the $\Delta F$ value or the recruitment threshold from the Control condition. The significance level was set at $P < 0.05$, and all values are reported as means $\pm$ SD.

## Results

### Pain

Participants reported an average pain intensity of $4.6 \pm 1.2$ out of 10 before the trapezoidal contractions and $4.0 \pm 1.3$ out of 10 before the triangular contractions. During contractions, the reported pain levels were $3.3 \pm 2.2$ for the trapezoidal contractions and $4.0 \pm 1.6$ for the triangular contractions. Pain was mainly localized around the injection site.

### Force and interference electromyography

The effective force ($Fz$), displayed as feedback and averaged across the three plateaus of the trapezoidal contractions, did not significantly differ between conditions (main effect of Condition: $F(2, 28) = 3.0$, $P = 0.066$), as further supported by the very low coefficient of variation across conditions ($1.2\% \pm 0.9\%$). Similarly, the other force components did not change significantly during Pain ($Fx$: $F(2, 28) = 0.8$, $P = 0.47$; $Fy$: $F(2, 28) = 4.3$, $P = 0.024$ with *post hoc* tests revealing no significant differences (all $P > 0.058$)). Of note, the correlation between effective force and the smoothed cumulative spike train did not differ significantly across conditions ($P = 0.865$; $0.59 \pm 0.16$ for Control, $0.58 \pm 0.13$ for Pain and $0.60 \pm 0.14$ for Washout).

The mean EMG amplitude calculated over the four grids did not significantly differ between conditions ($F(2,28) = 0.91$, $P = 0.42$).

## Motor unit yield

An average of $28.6 \pm 14.8$ motor units (range: 7–53) and $22.3 \pm 11.3$ motor units (range: 5–42) were identified per condition and participant during the trapezoidal and triangular contractions, respectively. Across the three conditions (Control, Pain and Washout), we successfully tracked 288 motor units for the trapezoidal contractions and 145 motor units for the triangular contractions. This resulted in an average of $19.2 \pm 13.3$ (range: 5–41)

motor units per participant for the trapezoidal contractions and $9.7 \pm 6.9$ (range: 1–22) motor units per participant for the triangular contractions. All results presented below are based on these tracked motor units. The entire data set is available at https://doi.org/10.6084/m9.figshare.28342568.v1

## Motor unit discharge characteristics

**Discharge rate.** When considering the mean discharge rate across the three plateaus of the trapezoidal contractions, there was a main effect of condition ($F(2, 856) = 8.4$, $P < 0.001$; Fig. 2). Specifically, the discharge rate was lower during Pain ($8.9 \pm 1.6$ pps) than both

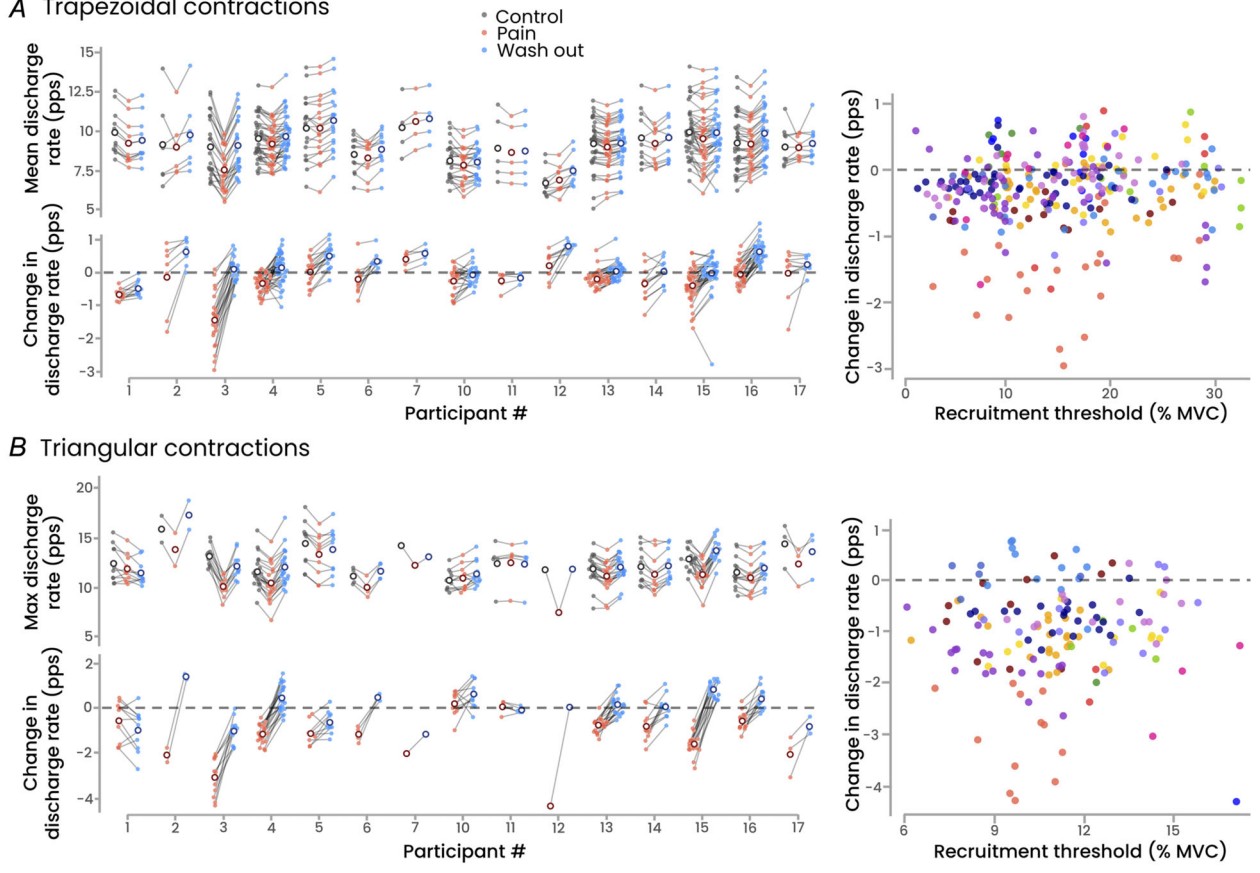

**Figure 2. Change in discharge rate during pain**
*A*, the discharge rate averaged over the plateau of the trapezoidal contractions for each participant (*n* = 15) and condition (Control, Pain and Washout). *B*, the maximal discharge rate measured during the triangular contraction for each participant (*n* = 15) and condition. In both panels, the lower figure shows the change in discharge rate relative to the Control condition, and the right figure shows the relationship between the change in discharge rate during Pain and the recruitment threshold assessed during the Control condition. In the left panels, each dot represents an individual motor unit and only motor units successfully tracked across all three conditions are included. The unfilled circles represent the mean values. In the right panels, each dot represents an individual motor unit and each colour represents a participant. Regardless of the task, there was a significant main effect of condition. The discharge rate was significantly lower during Pain compared with both Control and Washout, with no significant difference between Control and Washout. When considering the population level, there was no significant correlation between the change in discharge rate during Pain relative to Control and the recruitment threshold. [Colour figure can be viewed at wileyonlinelibrary.com]

Control (9.3 ± 1.7 pps; $P = 0.02$) and Washout (9.4 ± 1.6 pps; $P < 0.001$), with no significant difference between Control and Washout ($P = 0.36$). The change in discharge rate between Control and Pain was not significantly correlated with the pain level reported during contraction ($r = -0.45$; $P = 0.091$).

When considering the maximal discharge rate during the triangular contractions, there was also a main effect of condition ($F(2, 430) = 26.0$, $P < 0.001$; Fig. 2). Specifically, the discharge rate was significantly lower during Pain (11.1 ± 1.7 pps) than both Control (12.2 ± 1.8 pps; $P < 0.001$) and Washout (12.3 ± 1.7 pps; $P < 0001$), with no significant difference between Control and Washout ($P = 0.83$).

Despite the significant decrease in motor unit discharge rate during Pain, a notable proportion of motor units did not exhibit such a decrease, i.e. 24.8% of motor units during trapezoidal contractions (Fig. 2), with no significant correlation with the reported pain level during contraction ($r = -0.27$; $P = 0.32$). This proportion of motor units showing no decrease in discharge rate was 14.8% during triangular contractions.

It is worth noting that when considering the group level, there was no significant correlation between the recruitment threshold measured during the Control condition and the change in discharge rate between Control and Pain, regardless of the task (trapezoidal: $r = 0.06$; $P = 0.27$; triangular: $r = 0.02$; $P = 0.81$; Fig. 2). We also conducted this analysis for each individual participant and found no significant correlation in 11 out of 15 participants for the trapezoidal contractions ($P$-values ranging from 0.065 to 0.797) and in 11 out of 12 participants for the triangular contractions ($P$-values ranging from 0.140 to 0.934, with three participants excluded due to a low motor unit yield). Over the remaining participants who exhibited a significant correlation, one participant exhibited a negative correlation during both the trapezoidal ($r = -0.52$, $P < 0.001$) and the triangular contractions ($r = -0.72$, $P < 0.001$); and three participants exhibited a positive correlation ($r$ ranging from 0.42 to 0.68; $P$-values ranging from 0.004 to 0.038).

**Estimates of neuromodulation.** Because we were unable to find any pairs of control/test units in three participants (including the two female participants), $\Delta F$ values are reported for 12 participants. There was a main effect of condition ($F(2, 322) = 5.9$, $P = 0.003$; Fig. 3), with significantly lower $\Delta F$ values during Pain (2.2 ± 1.3 pps) compared with Control (2.7 ± 1.3 pps; $P = 0.002$), corresponding to a decrease of 11.8 ± 38.7% during Pain relative to Control. There was neither a significant difference between Pain and Washout (2.5 ± 1.1 pps; $P = 0.11$), nor between Control and Washout ($P = 0.32$).

Interestingly, there was a significant positive correlation between the change in maximal discharge rate between Control and Pain and the change in $\Delta F$ ($r = 0.61$, $P = 0.035$).

Using the geometric approach, which enabled us to isolate the effects of neuromodulation from inhibition (Beauchamp et al., 2023; Chardon et al., 2023), we found no significant main effect of condition, regardless of the metric considered: acceleration ($F(2, 430) = 2.2$, $P = 0.11$), attenuation ($F(2, 430) = 0.37$, $P = 0.69$) and brace height ($F(2, 430) = 1.91$, $P = 0.15$) (Fig. 3).

## Common synaptic input

We used two different approaches to estimate the common synaptic inputs received by the motor neuron pool during the trapezoidal contractions. Because less than eight motor units were identified in three participants, these analyses were performed on the remaining 12 participants.

There was a main effect of condition on the estimated proportion of common synaptic input ($F(2, 22) = 4.8$; $P = 0.019$). Specifically, the proportion of common input was significantly lower during Pain than both Control ($-12.6 ± 12.7\%$; $P = 0.031$) and Washout ($-9.1 ± 15.4\%$; $P = 0.040$), with no significant difference between Control and Washout ($P = 0.99$) (Fig. 4).

We also applied a principal component analysis to the smoothed discharge rates over the plateaus of the trapezoidal contractions and we quantified the variance explained by the first principal component. There was a significant main effect of condition ($F(2, 22) = 6.0$; $P = 0.008$), with the variance of the first component being lower during Pain compared with both Control ($-11.8 ± 14.2\%$; $P = 0.041$) and Washout ($-12.7 ± 18.6\%$; $P = 0.009$). No significant difference was observed between Control and Washout ($P = 0.77$) (Fig. 4).

## Simulations

The main outcomes of the simulations are depicted in Fig. 5. The six simulation scenarios produced realistic motor unit discharge rates, ranging from 10.4 pps under homogeneous inhibition to 13.0 pps in the Control scenario. Consistent with our experimental data, all simulation scenarios resulted in an overall decrease in discharge rates of the matched motor units, with a proportion of motor units exhibiting an increase in discharge rate to maintain force (ranging from 20.0 ± 0.4% to 39.5 ± 0.4% across all scenarios). In addition, some new motor units were recruited in the presence of inhibition, ranging from +1.7 ± 1.0 (Clustered inhibition) to +6.7 ± 0.7 (Gradient inhibition) motor units.

Compared with the Control scenario, the change in discharge rate across conditions varied from $-1.4 \pm 0.0$ pps for Gradient inhibition to $-0.5 \pm 0.1$ pps for Heterogeneous inhibition with high variance. When considering the estimates of common synaptic input, only three simulation scenarios aligned with our experimental data, showing a decrease in both the proportion of common input and the variance of the first principal component: the Gradient, Clustered and Heterogeneous (high variance) distributions of inhibition (Fig. 5). However, among these three scenarios, only two (Heterogeneous (high variance) and Clustered distribution) replicated the absence or weak relationship between the change in discharge rate and the recruitment threshold of the Control condition observed in our experimental data, with a mean $R^2 < 0.17$ (Fig. 5). Notably, the Gradient distribution logically resulted in a very high $R^2$ value indicating a strong relationship between the change in discharge rate and the recruitment

threshold. Overall, these simulation results demonstrate that only the two scenarios with high variability in the distribution of inhibitory inputs (Clustered and Heterogeneous (high variance) inhibition) successfully reproduced key features of our experimental data, i.e. an overall decrease in discharge rate regardless of motor unit size and a decrease in indexes of common synaptic input.

## Discussion

We investigated the behaviour of large samples of motor units during submaximal contractions at 30% of MVC under nociceptive stimulation to identify inhibitory patterns within the population of active motor units. Our experimental data, combined with *in silico* modelling, demonstrate an increase in inhibition during such moderate-intensity contractions, which is non-homogeneously distributed across motor units, with evidence indicating a lack of strong correlation between

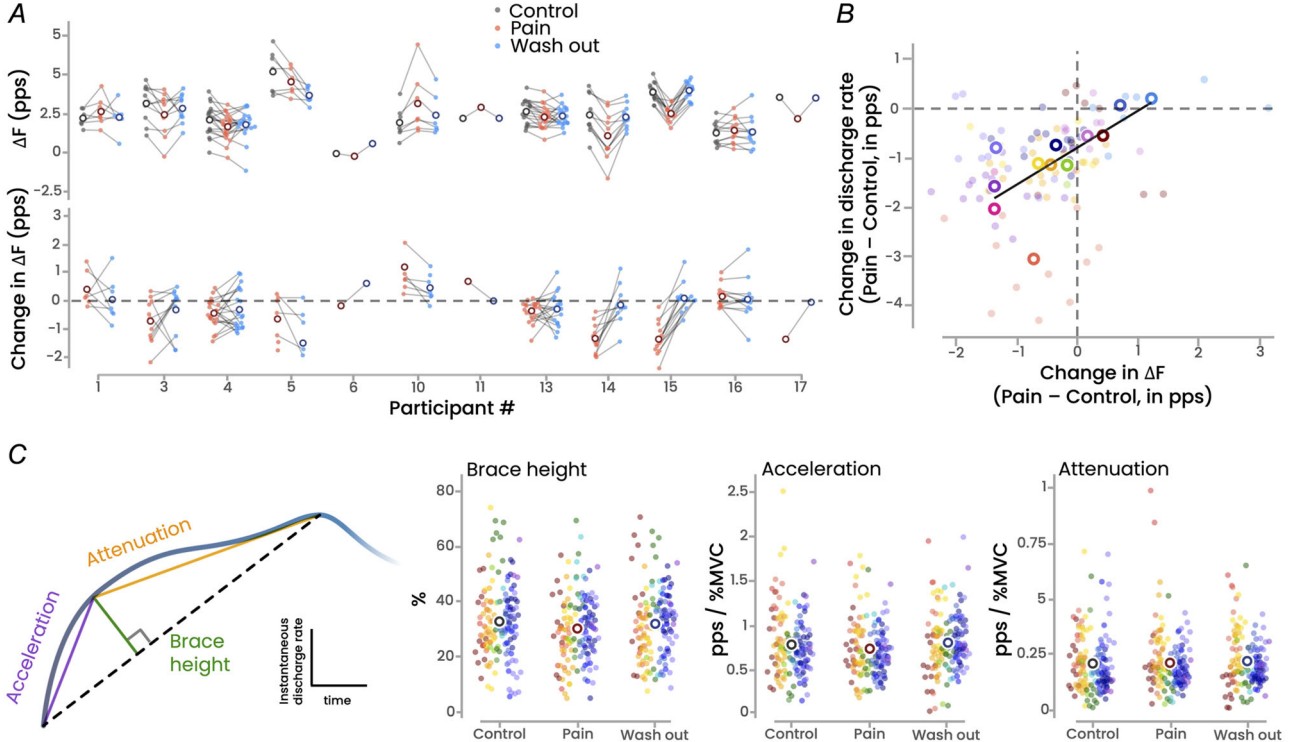

**Figure 3. Neuromodulatory input**

*A*, the $\Delta F$ values for each participant and condition (Control, Pain and Washout), with the lower portion showing the change in $\Delta F$ during Pain relative to Control ($n = 12$). There was a main effect of condition, with significant lower $\Delta F$ values during Pain compared with Control, with no other significant differences. Note the absence of data for three participants who were excluded due to a low motor unit yield. *B*, the significant positive correlation between the change in $\Delta F$ during Pain relative to Control and the change in maximal discharge rate measured during the triangular contraction. Each dot represents a motor unit, and each colour represents a participant, with the mean for each participant represented by an unfilled circle ($n = 12$). *C*, the geometric approach used to estimate the strength of neuromodulatory inputs. Results are presented on three distinct graphs where each dot represents a motor unit, and each colour represents a participant ($n = 15$). The group mean is represented by an unfilled circle. There was no significant main effect of condition, regardless of the metrics. [Colour figure can be viewed at wileyonlinelibrary.com]

the distribution of this inhibition and motor unit size. These findings not only enhance our understanding of the wide range of motor responses observed during pain but also provide a framework for refining contemporary theories on how movement control is altered in response to pain.

The 'pain adaptation' theory (Lund et al., 1991) proposes a decreased activation of the agonist muscles to limit the amplitude or velocity of the painful movements. This theory aligns with the reduction in motor unit discharge rate observed during contractions with matched force in the presence of noxious stimulation (Sanderson et al., 2021). However, since discharge rate is a key determinant of muscle force, other changes in motor output must occur to maintain force during pain, such as an increased discharge rate in some motor units or the recruitment of additional units (Hodges & Tucker, 2011). This aligns with emerging evidence showing non-homogeneous effects of noxious stimulation across motor units (Hodges & Tucker, 2011; Hug et al., 2013; Martinez-Valdes et al., 2020; Tucker et al., 2009). Our results support these findings, showing that despite an overall significant decrease in discharge rate, a proportion of motor units did not show this decrease, and some even exhibited an increase in discharge rate (24.8% of motor units during trapezoidal contractions; Fig. 2). Our aim was to gain deeper insights into the neural mechanisms underlying these variable responses within the motor unit pool.

Changes in motor unit discharge rate during pain can result from changes in the net synaptic input to motor neurons or from changes in motor neuron excitability driven by neuromodulatory inputs. Neuromodulatory drive modulates persistent inward currents in motor neuron dendrites, amplifying excitatory depolarizing currents and facilitating motor unit discharge at lower excitatory input levels than those required for recruitment,

a phenomenon known as discharge rate hysteresis (Lee & Heckman, 1998). In our study, we used a combination of approaches to assess these different effects of persistent inward currents (Beauchamp et al., 2023). We observed a significant decrease in $\Delta F$, an index of discharge rate hysteresis, which correlated with the decrease in discharge rate (Fig. 3). However, because $\Delta F$ is largely influenced by inhibitory inputs (Beauchamp et al., 2023; Chardon et al., 2023; Revill & Fuglevand, 2017), this change cannot be confidently attributed to a change in neuromodulatory drive. To address this limitation, we used complementary metrics that assess the deviations from a linear increase in discharge rate during the ramp-up phase of the triangular contractions (acceleration, brace height and attenuation) (Beauchamp et al., 2023). None of these metrics were significantly modified during Pain. While it does not definitely rule out an effect of pain on neuromodulation, it is noteworthy that these metrics were derived from a large sample of motor units ($n = 145$), suggesting that neuromodulation is unlikely to be a major factor in the observed changes in motor unit discharge rate during transient experimental pain. However, it requires confirmation on a larger sample size. At first glance, this may appear inconsistent with the role of mono-amine pathways in pain control (Martins & Tavares, 2017), which could increase persistent inward currents through monoaminergic release at motor neurons. However, this mechanism has been predominantly observed in chronic pain conditions (Bannister et al., 2009), and it is possible that transient pain may not be sufficient to elicit such a pain control mechanism.

As the neuromodulatory drive was not significantly altered in the presence of pain, the overall decrease in motor unit discharge rate likely results primarily from changes in net synaptic inputs to motor neurons. Although this could theoretically be explained by a

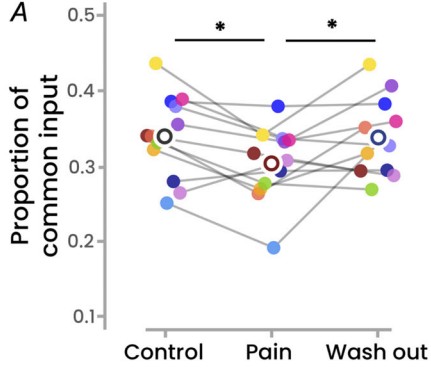
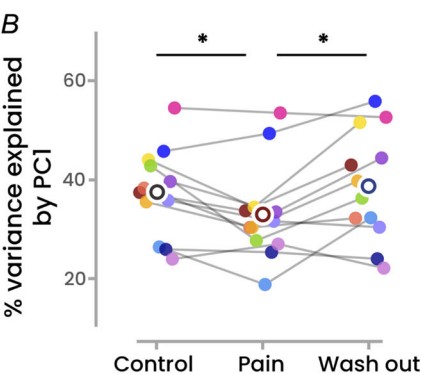

**Figure 4. Estimates of common synaptic inputs**
Two different approaches were used to estimate the common synaptic inputs received by the motor neuron pool during the trapezoidal contractions, that is, the proportion of common synaptic input and the variance explained by the first principal component. Regardless of the metric, there was a main effect of condition, with significant lower values during Pain compared with Control. Asterisks indicate significant differences ($P < 0.005$). Each dot represents a participant ($n = 12$), and the unfilled circles represent the mean values. PC1, first principal component. [Colour figure can be viewed at wileyonlinelibrary.com]

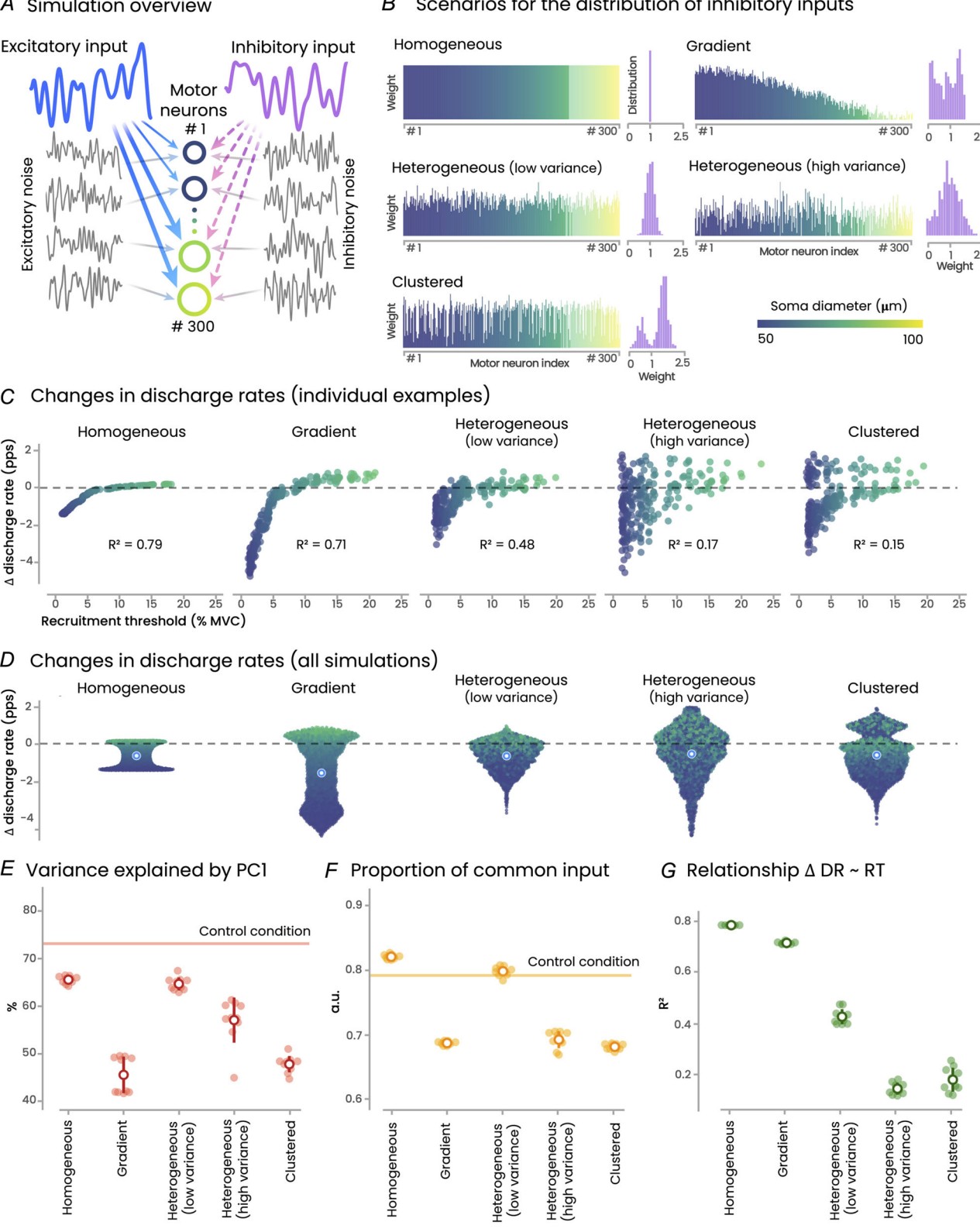

**Figure 5. *In silico* model**

We simulated the behaviour of 300 motor units using a conductance-based leaky integrate-and-fire model. *A*, the structure of the simulation model (see *Methods* for further details). *B*, the five simulated scenarios, distinguished by different distributions of inhibitory inputs. The weights of the inhibitory input were distributed as follows: (1) uniformly (homogeneous), (2) biased toward the low-threshold units (gradient), (3) heterogeneously with either

low or high variance (heterogeneous), and (4) distributed into two groups, one receiving low inhibition and the other high inhibition. Each scenario was run 10 times and compared with a Control scenario with no inhibition. *C*, individual examples of the relationship between the change in discharge rate compared with Control and the recruitment threshold for each recruited motor unit. *D*, the results from the 10 simulations, with each dot representing a motor unit and each unfilled circle indicating the mean. *E* and *F* show the two metrics used to estimate the proportion of common synaptic input, and *G* shows the coefficient of determination for the relationship between the change in discharge rate relative to Control and recruitment threshold in the Control condition. In panels *E*, *F* and *G*, each dot corresponds to the result of a simulation ($n = 10$). PC1, first principal component; RT, recruitment threshold; DR, discharge rate; MVC, maximal voluntary contraction. [Colour figure can be viewed at wileyonlinelibrary.com]

reduction in excitatory drive, it is unlikely given that the net joint torque and muscle activation level (assessed through EMG amplitude) remained unchanged across conditions. In addition, a reduction in excitatory cortico-spinal drive would likely have affected all units similarly, which does not align with our results. Therefore, increased inhibitory input is the most likely explanation for our experimental results. The inhibitory effects of group III and IV nociceptive afferents, which are activated by hypertonic saline, likely contribute to the reduced motor unit discharge rate during contractions with noxious stimulation (Mense, 1993). While this aligns with the overall decrease in discharge rate observed in our study, it does not account for the proportion of motor units that did not exhibit a decrease (Fig. 2). This observation is consistent with previous reports of non-homogeneous effects of noxious stimulation within the motor unit pool. For instance, Martinez-Valdes et al. (2020) observed a divergent response between low- and high-threshold motor units during high-intensity contractions (70% of MVC), with high-threshold units increasing their discharge rate during pain, while the discharge rate of lower-threshold remained unchanged. As elaborated in the introduction, these results do not necessarily indicate non-homogeneous inhibition across the motor unit pool, as they could result from a homogeneous inhibition that disproportionately affects low-threshold units. This possibility is demonstrated by our simulation results (Homogeneous inhibition, Fig. 5).

To gain deeper insights into the distribution of inhibitory inputs from our large samples of motor units, we used *in silico* models to determine the inhibition schemes compatible with our experimental results. It is important to note that under the constraint of constant force output, even the Homogeneous inhibition scenario – where all the motor units received the same level of inhibitory input – could reproduce heterogeneous motor unit behaviour, with low-threshold units showing decreased discharge rates and higher-threshold units showing increased discharge rates. However, to validate a simulation model as a plausible inhibition scheme, it needed to replicate all four key experimental observations: (i) an overall decrease in discharge rate, (ii) a reduced variance explained by the first principal component, (iii) a reduced proportion of common input, and (iv) no

(or weak) relationship between the change in discharge rate and the recruitment threshold. Only two simulation scenarios met these criteria: Clustered inhibition and Heterogenous inhibition with high variance. These simulation findings strongly support that, during moderate-intensity contractions, inhibitory inputs are non-homogeneously distributed among motor units, independent of their size. This may explain why newly recruited higher-threshold units during pain do not always follow the expected orderly recruitment pattern (Tucker et al., 2009), a phenomenon that our simulations also revealed in the Heterogeneous or Clustered inhibition scenarios.

Although the neural basis for this non-homogeneous inhibition across the pool is not yet fully understood, several potential mechanisms can be proposed. One possibility is a differential effect of small-diameter group III and IV muscle afferents on motor neurons, with either a facilitatory or inhibitory effect, as previously suggested (Kniffki et al., 1981; Martin et al., 2006). Another possibility is that motor units within the same pool receive distinct feedback, as suggested by studies on *Drosophila*, where tibia flexor motor neurons receive distinct feedback signals from leg proprioceptors (Azevedo et al., 2020). This pattern of non-homogeneous inhibition also aligns with theories suggesting that motor units within the same pool may receive different net inputs and thus be represented in different synergies (Hug et al., 2023). This concept is particularly well illustrated in the vastus lateralis muscle (the muscle studied here), where the activity of motor units is represented by multiple underlying patterns (latent factors) (Del Vecchio et al., 2023; Dernoncourt et al., 2025). This suggests that the non-homogeneous distribution of group III and IV afferent feedback may shape the dimensionality of this manifold, consistent with the influence of subcortical pathways, including recurrent inhibition (Dernoncourt et al., 2025). Interestingly, a decrease in the proportion of common input has also been reported in the vastus lateralis muscle of individuals who underwent anterior cruciate ligament reconstruction (Nuccio et al., 2024). The purpose of such a non-homogeneous inhibition remains an open question. It may serve as a cost-effective strategy to maintain force while limiting the recruitment of additional large motor units, which have a higher

metabolic cost (Zierath & Hawley, 2004). However, this hypothesis still needs to be tested.

Several methodological issues need consideration. First, nociceptive stimulation was applied to a non-muscular tissue to ensure that hypertonic saline affected only small-diameter nociceptive fibres, avoiding any impact on other axons, including motor neurons (Tucker & Hodges, 2009). However, we did not use an isotonic saline bolus injection as an additional Control condition. This is because it would have extended the protocol and increased the risk of fatigue; and based on previous reports there is no effect of isotonic saline on motor unit discharge rates (Martinez-Valdes et al., 2020, 2021). We are confident that the observed effects were due to the painful stimulus, and not the small addition of fluid within the knee fat pad. Second, to further avoid fatigue and because $\Delta F$ validation for estimating the magnitude of persistent inward currents is not established at high force levels, our protocol was limited to contractions at 30% of MVC. Even though our results need to be confirmed across a broader range of forces, we believe that this limitation does not impact our main conclusion that a noxious stimulus induces non-homogeneous inhibition within a group of motor units with varying recruitment thresholds. Third, while EMG decomposition allowed us to identify a large sample of motor units, it cannot support conclusive outcomes regarding motor unit recruitment and de-recruitment during pain. Specifically, the inability to track a motor unit across conditions should not be directly interpreted as a change in recruitment, as it may result from the decomposition algorithm failing to identify the same motor unit between conditions. Fourth, we acknowledge that our simulation model simplifies representation of the complex biophysical processes underlying motor unit behaviour. This simplification likely accounts for certain discrepancies between simulation and experimental results (e.g. higher values of proportion of common inputs for the simulation results, Fig. 5*E*). Nevertheless, the model successfully reproduced the key experimental outcomes, including realistic discharge rates, the decrease in discharge rate due to inhibitory inputs, and the change in proportion of common input. Finally, and importantly, these results need to be confirmed in females as we were unable to identify enough motor units in females for estimating $\Delta F$ and common input, a known limitation of surface EMG decomposition (Taylor et al., 2022).

To conclude, this study provides novel insights into the distribution of inhibitory inputs to motor units during moderate-intensity contractions performed in the presence of experimental pain. Specifically, we found that pain does not significantly alter the neuromodulatory drive to motor neurons, suggesting that the observed changes in motor unit behaviour are primarily driven by changes in inhibitory inputs. Our combined experimental and *in silico* modelling approach revealed that these inhibitory inputs are non-homogenously distributed across motor units, independent of motor unit size. These findings challenge traditional views of homogeneous inhibition during pain and provide a framework for refining contemporary theories on how movement control is affected by pain. Further work is needed to clarify the functional implications of this response, particularly regarding its impact on the metabolic cost of contraction and the control of force.

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

## Additional information

### Data availability statement

The entire dataset (raw and processed data) is available at: https://doi.org/10.6084/m9.figshare.28342568.v1; and the *in silico* model is available at: https://github.com/FrancoisDernoncourt/Pain_inhibition_simulation

### Competing interests

The authors declare no competing financial interests.

### Author contributions

Contribution and design of the experiment: F.H., J.T., M.B., K.T.; Collection of data: F.H., W.H., K.T.; Analysis: F.H., F.D., S.A.; Drafting the article or revising it for important intellectual content: F.H., F.D., S.A., J.T., M.B., W.H., K.T.; All authors approved the final version of the manuscript.

### Funding

François Hug is supported by the French government, through the UCAJEDI Investments in the Future project managed by the National Research Agency (ANR) with the reference number ANR-15-IDEX-01 and by an ANR grant (ANR-24-CE17-5805, NEUROMOTOR project).

### Keywords

common input, electromyography, motor neuron, persistent inward currents, principal component analysis

### Supporting information

Additional supporting information can be found online in the Supporting Information section at the end of the HTML view of the article. Supporting information files available:

**Peer Review History**

