## [Peer Review History · The Journal of Physiology]

Non-Homogeneous Distribution of Inhibitory Inputs Among Motor Units in Response to Nociceptive Stimulation at Moderate Contraction Intensity

Francois Hug, Francois Deroncourt, Simon Avrillon, Jacob Thorstensen, Manuela Besomi, Wolbert van den Hoorn, and Kylie J Tucker

DOI: 10.1113/JP288504

Corresponding author(s): Francois Hug (francois.hug@univ-cotedazur.fr)

Review Timeline:

Submission Date:	13-Jan-2025
Editorial Decision:	17-Mar-2025
Revision Received:	27-Mar-2025
Editorial Decision:	28-Apr-2025
Revision Received:	28-Apr-2025
Accepted:	12-May-2025

Senior Editor: Richard Carson

Reviewing Editor: Madeleine Lowery

Transaction Report:

Dear Dr Hug,

Re: JP-RP-2025-288504 "Heterogeneous distribution of inhibitory inputs among motor units as a key neural mechanism in the motor response to pain" by Francois Hug, Francois Dernoncourt, Simon Avrillon, Jacob Thorstensen, Manuela Besomi, Wolbert van den Hoorn, and Kylie J Tucker

Thank you for submitting your manuscript to The Journal of Physiology. It has been assessed by a Reviewing Editor and by 2 expert referees and we are pleased to tell you that it is potentially acceptable for publication following satisfactory major revision.

REVISION CHECKLIST:

We look forward to receiving your revised submission.

Yours sincerely,

Richard Carson
Senior Editor
The Journal of Physiology

REQUIRED ITEMS

- Author photo and profile. First or joint first authors are asked to provide a short biography (no more than 100 words for one author or 150 words in total for joint first authors) and a portrait photograph. These should be uploaded and clearly labelled together in a Word document with the revised version of the manuscript. See Information for Authors for further details.

- You must start the Methods section with a paragraph headed Ethical Approval. If experiments were conducted on humans, confirmation that informed consent was obtained, preferably in writing, that the studies conformed to the standards set by the latest revision of the Declaration of Helsinki and that the procedures were approved by a properly constituted ethics committee, which should be named, must be included in the article file. If the research study was registered (clause 35 of the Declaration of Helsinki), the registration database should be indicated, otherwise the lack of registration should be noted as an exception (e.g. The study conformed to the standards set by the Declaration of Helsinki, except for registration in a database). For further information see: <https://physoc.onlinelibrary.wiley.com/hub/human-experiments>.

- Your manuscript must include a complete Additional Information section, including competing interests; funding; author contributions and acknowledgements.

- The Journal of Physiology funds authors of provisionally accepted papers to use the premium BioRender site to create high resolution schematic figures. Follow this link and enter your details and the manuscript number to create and download figures. Upload these as the figure files for your revised submission. If you choose not to take up this offer, we require figures to be of similar quality and resolution. If you are opting out of this service to authors, state this in the Comments section on the Detailed Information page of the submission form. The link provided should only be used for the purposes of this submission. Authors will be charged for figures created on this premium BioRender account if they are not related to this manuscript submission.

- Please ensure that the Article File you upload is a Word file.

- Papers must comply with the Statistics Policy: https://jp.msubmit.net/cgi-bin/main.plex?form_type=display_requirements#statistics.

In summary:

- If n {less than or equal to} 30, all data points must be plotted in the figure in a way that reveals their range and distribution. A bar graph with data points overlaid, a box and whisker plot or a violin plot (preferably with data points included) are

acceptable formats.

- If $n > 30$, then the entire raw dataset must be made available either as supporting information, or hosted on a not-for-profit repository, e.g. FigShare, with access details provided in the manuscript.
- 'n' clearly defined (e.g. x cells from y slices in z animals) in the Methods. Authors should be mindful of pseudoreplication.
- All relevant 'n' values must be clearly stated in the main text, figures and tables.
- The most appropriate summary statistic (e.g. mean or median and standard deviation) must be used. Standard Error of the Mean (SEM) alone is not permitted.
- Exact p values must be stated. Authors must not use 'greater than' or 'less than'. Exact p values must be stated to three significant figures even when 'no statistical significance' is claimed.

EDITOR COMMENTS

Reviewing Editor:

This study examines the contribution of inhibitory and neuromodulatory inputs to motor unit firing in the vastus lateralis muscle in the presence of pain. Both reviewers appreciate the experimental methodology and the mechanistic insights provided. They have raised a number of important questions regarding the interpretation of results and points requiring clarification which should be addressed.

Please also see 'Required Items' above.

REFEREE COMMENTS

Referee #1:

See attached file.

Referee #2:

The present study aimed to examine the distribution pattern of inhibitory inputs to the vastus lateralis motor unit pool in response to pain during low/moderate knee-extension contraction. The article is well-written, and the experiments were conducted with rigor, incorporating multiple insightful analyses. The findings provide novel evidence of the heterogeneous distribution of nociceptive inputs to a pool of lower-threshold motor units. While these results are compelling, there are some areas that require the authors' attention.

My primary concern with this study is the force level assessed, and the strong conclusions drawn despite only evaluating a single force level. While previous studies demonstrating reductions in discharge rate with pain have been conducted at similar force levels (10%-30% MVC), assumptions regarding differential inhibition across the motor unit pool (i.e., lower versus higher threshold motor units) have been based on much higher force levels (~70% MVC). In the present study, the authors examined the quadriceps, a muscle with a high upper recruitment limit (approaching 100% MVC). As a result, the assumptions made here primarily apply to lower-threshold motor units, rather than the full motor unit pool. This distinction should be clearly reflected in the title and throughout the manuscript. Although the results remain valuable, this study does not fully address heterogeneous inhibitory (or excitatory) inputs across the entire motor unit pool, particularly for higher-threshold motor units, which were not assessed. Additionally, several relevant studies on changes in common synaptic inputs should be incorporated into the discussion. Finally, some additional analyses (outlined below) could further strengthen the study's implications.

Other minor comments below:

Please add line numbers

Inform force intensity assessed in the abstract

One important aspect that is completely overlooked in the introduction is that the effects of pain on motor unit firing properties are both force- and velocity-dependent. The authors present previous evidence as if studies report conflicting responses, but they should also acknowledge that these differences may be influenced by variations in force intensity and task demands. For example, Martinez-Valdes et al. (2020) observed reductions in discharge rate at low forces (consistent with many previous studies) but reported maintained or even increased discharge rates at higher force levels. The authors should discuss these variations across studies and consider how force intensity and task-specific factors might influence the motor unit firing properties assessed in response to pain.

Input to simulated neurons: 'about 12 N.m for the number of simulated motor units.' - Does this refer to motor unit twitch force? Or Knee extension torque? clarify.

In my opinion, the hypothesis tested by Martinez-Valdes et al. 2020, cannot be tested at 30% MVC. A large proportion of the MU pool of the VL is unfortunately not assessed at this force level.

Pain: What was the pain duration? Did the pain intensity vary significantly while recording the multiple contractions? Did pain intensity have an effect on the motor unit properties assessed in the present study?

Change to interference EMG

'The mean EMG amplitude calculated over the grid' - wasn't this calculated for all the 4 grids?

Any association between the level of heterogeneity in inputs and pain level experienced? - in other words, would the participants experiencing more pain show higher or lower heterogeneity? This would help understand the functional implication of the findings.

Discussion

A key finding of this study is that, despite the overall reduction in firing rate in response to experimental pain, there appears to be a heterogeneous distribution of inhibitory and excitatory inputs across the motor unit pool. However, the authors should emphasize that these changes are primarily observed in lower-threshold motor units. Comparisons with Martinez-Valdes et al. (2020) are somewhat challenging, as that study also reported a similar reduction in discharge rate at 20% MVC but divergent behaviors at 70% MVC. However, unlike the present study, it did not assess the differential distribution of inputs across the pool of lower threshold motor units. Therefore, I believe the current study provides several additional insightful analyses that contribute meaningfully to this area of research, but claims need to be more specific to the population of units assessed.

Regarding common input, it is surprising that the studies by Farina et al. (J Neurophysiol, 2011) and Yavuz et al. (J Neurophysiol, 2015) are not included among the references. These studies reported findings that contrast with those

presented here and should be discussed in relation to the current results. In particular, PCI primarily reflects changes in low-frequency coherence (<5 Hz), whereas Farina et al. (2011) found an increase in this frequency range in response to pain. It is also unexpected that, despite no observed variations in torque steadiness, the authors report a significant decline in PCI and a reduction in the variance explained by PC1. What mechanisms might underlie these differences? Additionally, do the authors observe changes in CST versus torque cross-correlation? Examining this relationship could provide insight into potential compensatory strategies employed during task execution.

'Martinez-Valdes et al. (2020) observed a divergent response between low- and high-threshold units' - Emphasise that this was only found at high forces, at low forces (20% MVC) Martinez-Valdes et al., 2020 found the same reductions in discharge rate as the ones reported in the present study.

The assumptions made by Mesquita et al. 2020, also consider higher forces, therefore, the PIC findings of the present study do not fully explain the potential differential inhibition/excitation among lower and higher threshold units.

'These simulation findings strongly support a non-homogeneous distribution of inhibitory inputs to motor units, independent of their size' - Yes, but only at low to moderate force levels. What happens at higher forces remains unknown.

"This may explain why newly recruited higher-threshold units during pain do not always follow the expected orderly recruitment pattern (Tucker et al., 2009)." Were the authors able to identify newly recruited motor units in response to pain? To date, Tucker et al. (2009) remain the only study to report such changes, but their findings were based on more limited methodologies. Given the larger motor unit samples obtained in the present study, it would be valuable to assess whether the number of identified units differs across conditions. Specifically, were unique motor units recruited in the pain condition? Did you see a violation of the size principle? Analyzing this aspect could provide further insights into the recruitment strategies underlying motor unit behavior in response to pain.

END OF COMMENTS

Heterogeneous distribution of inhibitory inputs among motor units as a key neural mechanism in the motor response to pain. Hug et al.

This is an interesting and well written paper. I found the methodologically employed to be rigorous. I welcomed the combination of experimental findings and modelling. I also appreciated the attention paid to the motor units that behaved differently from the group averages.

I have a few specific comments for the authors to consider:

Pg 5, para 1: Can the authors justify only sampling from 2 females? Because the two females were excluded from the delta F analysis, that analysis was performed strictly on male participants. Sex differences in motor control are known. Paper after paper, the limitations of the EMG decomposition technique for extracting motor units from female participants is articulated. I realize that the authors did not develop the software but continuing to focus primarily on male participants is not acceptable. Could the authors go back to intramuscular fine wire recordings in females to confirm the results? I realize the sample size would be smaller but currently the sample size is zero for females for the delta F analysis.

Pg 6, para 1: Why did participants perform a knee flexion contraction before the ramp?

Page 6: Why did the authors choose vastus lateralis?

Pg 7, last para: How are the authors sure the manual editing did not introduce bias into the MU spike trains?

Pg 8, para 3: "A motor unit was considered tracked if at least 30% of the discharge times were shared". A threshold of 30% seems low to me. Why was template matching not used?

Pg 15, para 1: Why did the reported pain lessen during trapezoidal contractions but not triangular contractions?

Pg 15, para 3: "The entire data set is available at <https://figshare.com/s/8b978f1cd32d8329266e>." The link did not work for me.

Pg 18, last para: "Because less than eight motor units were identified in three participants, these analyses were performed on the remaining 12 participants." Did the excluded participants include both females like the delta F analysis?

Generally, some of the p values are between 0.6 and 0.1 suggesting a failure to power the study sufficiently. What are the effect sizes? How did the authors determine the sample size a priori?

Pg 20: In Figure 4A, the proportion of common input is about 0.3. In the simulated data, Figure 5F, the proportion of common input is around 0.7 or above, even in the clustered and heterogeneous high variance conditions. You state that the simulation needs to match all four experimental results, but I don't think any of the models reproduced the PCI data. So how well is the model mimicking the physiology? This should be discussed.

Pg 20: Figure 5D: What is the x axis on this Figure?

Pg 24, last para: "by multiple latent factors defining a multidimensional manifold." Could this sentence be written with less jargon initially and then expanded upon with the technical

language? This would improve the understanding for the reader without expertise in that technique.

Point-by-point responses to the reviewers

We thank the reviewers for their constructive feedback. In response to their comments, we have revised the manuscript and provided detailed responses to each point. We believe that these revisions have strengthened the manuscript.

Referee #1

This is an interesting and well written paper. I found the methodologically employed to be rigorous. I welcomed the combination of experimental findings and modelling. I also appreciated the attention paid to the motor units that behaved differently from the group averages.

We thank the reviewer for their thorough review and insightful comments, which guided our edits and have strengthened the manuscript.

I have a few specific comments for the authors to consider:

Pg 5, para 1: Can the authors justify only sampling from 2 females? Because the two females were excluded from the delta F analysis, that analysis was performed strictly on male participants. Sex differences in motor control are known. Paper after paper, the limitations of the EMG decomposition technique for extracting motor units from female participants is articulated. I realize that the authors did not develop the software but continuing to focus primarily on male participants is not acceptable. Could the authors go back to intramuscular fine wire recordings in females to confirm the results? I realize the sample size would be smaller but currently the sample size is zero for females for the delta F analysis.

The reviewer is entirely right about the significant issue of female underrepresentation in motor unit studies, which indeed stems from the challenges in identifying sufficient number of motor units using surface EMG decomposition methods. The outcomes we present follow many attempts from our labs, both in France and Australia, to identify motor units from EMG decomposition in females across many different muscle groups. We have tried to collect this data from women with incredibly low subcutaneous fat layers, and with very high levels of aerobic training.

For the current manuscript, as the reviewer would expect, we aimed to recruit an equal number of male and female participants. However, following data collection from the first female participants, we observed that significantly fewer motor units could be identified in females, consistent with the literature. Given that our analyses (particularly the assessment of common input and ΔF) required large numbers of motor units, and to avoid unnecessary pain injections with limited motor unit yield, we subsequently focused recruitment solely on male participants.

Although we are unaware of previous intramuscular EMG studies reporting between-sex differences in motor unit response to experimental pain, we acknowledge that the unequal distribution of males and females is a limitation to the broad interpretation of our results across men and women. We have explicitly addressed this in the Discussion section: "*Finally, and importantly, these results need to be confirmed in females as we were unable to identify enough motor units in females for estimating ΔF and common input, a known limitation of surface EMG decomposition (Taylor et al., 2022).*"

Importantly, as our analyses required a large sample of motor units, conventional fine-wire electrodes - which detect only a few motor units at low contraction intensities - could not be used. High-density intramuscular thin-film electrodes might offer a better alternative than fine-wires; however, they are neither commercially available nor widely accessible (to our knowledge). Nevertheless, we fully recognise the importance to refine algorithms and/or surface electrode design to mitigate this limitation. We have secured funding for a project (link) using an innovative simulation framework that generates highly realistic EMG signals (Maksymenko et al., 2023). A part of this project aims to identify the key feature(s) that

explain between-sex differences in surface EMG decomposition outcomes, ultimately guiding improvements in electrode design and decomposition algorithms.

References:

- Maksymenko K, Clarke AK, Mendez Guerra I, Deslauriers-Gauthier S, Farina D. A myoelectric digital twin for fast and realistic modelling in deep learning. *Nat Commun.* 2023 Mar 23;14(1):1600.
- Taylor CA, Kopicko BH, Negro F & Thompson CK. (2022). Sex differences in the detection of motor unit action potentials identified using high-density surface electromyography. *J Electromyogr Kinesiol* 65, 102675.

Pg 6, para 1: Why did participants perform a knee flexion contraction before the ramp?

Participants performed a brief knee flexion (antagonist muscle contraction), as antagonist muscle contraction effectively reduces the amplitude of persistent inward currents (Gomes et al., 2024). The text has been revised as follows: *“To minimize the impact of the preceding contractions on persistent inward currents, a two-minute rest period was systematically provided before the triangular contractions. **At the beginning of this rest period, participants briefly contracted the antagonist muscles (knee flexion), as such contraction effectively attenuates persistent inward (Gomes et al., 2024) via reciprocal inhibition**”.*

Reference:

- Gomes MM, Jenz ST, Beauchamp JA, Negro F, Heckman CJ, Pearcey GEP. Voluntary co-contraction of ankle muscles alters motor unit discharge characteristics and reduces estimates of persistent inward currents. *J Physiol.* 2024;602(17):4237-4250.

Page 6: Why did the authors choose vastus lateralis?

We selected the vastus lateralis as the target muscle because this study is part of a larger research program investigating changes in motor control strategies in individuals with anterior knee pain. To record a large number of motor units, the vastus lateralis was more suitable than the vastus medialis, due to its larger size, which enabled the placement of 256 electrodes (i.e. four grids of 64 electrodes).

The following text has been added: *“To record a large sample of motor units, we selected the vastus lateralis (VL), which allowed the placement of four adhesive grids of 64 electrodes each (total of 256 electrodes; inter-electrode distance: 8 mm; GR08MM1305, OT Bioelettronica, Italy)”.*

Pg 7, last para: How are the authors sure the manual editing did not introduce bias into the MU spike trains?

Surface EMG decomposition has been extensively validated using both experimental and simulated signals (Holobar and Farina, 2014; Holobar et al., 2014). Manual editing is a necessary step to correct for obvious falsely identified discharge times and missed discharge times, as outlined by a recent expert consensus (Martinez-Valdes et al., 2023). In our study, we manually edited all the motor unit spike trains following previously published procedures (Del Vecchio et al., 2020; Hug et al., 2021). Importantly, this procedure has demonstrated high reliability across operators (Hug et al., 2021a) and has been validated against simulated signals (Rossato et al., 2024).

To address the reviewer’s comment the text has been revised as follows: *“Then, we manually edited all the motor unit spike trains following previously published procedures (Del Vecchio et al., 2020; Hug et al., 2021). **Of note, this procedure has demonstrated high reliability across operators (Hug et al., 2021a) and has been validated against simulated signals (Rossato et al., 2024).**”*

Importantly, to ensure transparency and reproducibility, the full data set - including raw EMG data, as well as unedited and edited spike trains - is publicly available in a data repository.

References:

- Holobar, A., Farina, D., 2014. Blind source identification from the multichannel surface electromyogram. *Physiol Meas* 35, R143-165.
- Holobar, A., Minetto, M.A., Farina, D., 2014. Accurate identification of motor unit discharge patterns from high-density surface EMG and validation with a novel signal-based performance metric. *J Neural Eng* 11, 016008.
- Martinez-Valdes E, Enoka RM, Holobar A, McGill K, Farina D, Besomi M, Hug F, Falla D, Carson RG, Clancy EA, Disselhorst-Klug C, van Dieën JH, Tucker K, Gandevia S, Lowery M, Søgaard K, Besier T, Merletti R, Kiernan MC, Rothwell JC, Perreault E, Hodges PW. Consensus for experimental design in electromyography (CEDE) project: Single motor unit matrix. *J Electromyogr Kinesiol.* 2023 Feb;68:102726. doi: 10.1016/j.jelekin.2022.102726

Pg 8, para 3: "A motor unit was considered tracked if at least 30% of the discharge times were shared". A threshold of 30% seems low to me. Why was template matching not used?

This approach has been used in previous studies based on the assumption that different motor units are unlikely to exhibit substantial synchronous firings at exactly the same time (e.g. Holobar et al., 2010; Avrillon et al., 2024). This method is more conservative compared to template matching, as matched motor units typically share nearly 100% of their spikes, while distinct motor units share only a very small proportion of spikes (typically less than 5%). A 30% threshold has been proposed to accommodate the use of unedited spike trains, which may contain false positive and negative spikes.

Template matching was not employed because it requires setting an arbitrary threshold for the correlation coefficient and frequently leads to ambiguous results where multiple motor unit pairs exceed this threshold, complicating clear discrimination. Importantly, when identifying duplicate motor units, i.e. the same motor unit detected across multiple electrode grids, the approach we used is the only suitable method, as the MUAP shapes of the same motor unit are different across grids.

References:

- Holobar A, Minetto MA, Botter A, Negro F, Farina D. Experimental analysis of accuracy in the identification of motor unit spike trains from high-density surface EMG. *IEEE Trans Neural Syst Rehabil Eng.* 2010
- Avrillon S, Hug F, Enoka RM, Caillet AHD, Farina D. The identification of extensive samples of motor units in human muscles reveals diverse effects of neuromodulatory inputs on the rate coding. *Elife.* 2024 Dec 9;13:RP97085.

Pg 15, para 1: Why did the reported pain lessen during trapezoidal contractions but not triangular contractions?

Thank you for raising this interesting question. The most likely explanation is that the triangular contraction was always performed first (during the Pain condition only), i.e., when pain intensity was at its highest. This choice was deliberate as performing the trapezoidal contractions first would have required a subsequent two-minute rest period before the triangular contractions (to minimize the impact of preceding contractions on persistent inward currents). Given that pain induced by hypertonic saline injection typically lasts < 5 min with a VAS score greater than 2, there was a risk that pain would subside during this resting period. Performing the triangular contraction first ensured sufficient time to complete one triangular (or occasionally two, if force matching was inaccurate) and three trapezoidal contractions under significant pain levels.

We have added this information to the manuscript: "***Because pain induced by hypertonic saline injection typically lasts less than 5 minutes (Martinez-Valdes et al., 2020), the triangular contraction was always performed first during the Pain condition. This sequence avoided an additional two-minute rest period required if trapezoidal contractions were performed first, thus ensuring that all contractions were performed while significant pain levels (greater than 2/10) were experienced.***"

Pg 15, para 3: "The entire data set is available at: <https://figshare.com/s/8b978f1cd32d8329266e>." The link did not work for me.

We apologize for this. The links to the data (raw and processed) and all code (including the simulation) have now been updated:

<https://doi.org/10.6084/m9.figshare.28342568.v1>

Pg 18, last para: “Because less than eight motor units were identified in three participants, these analyses were performed on the remaining 12 participants.” Did the excluded participants include both females like the delta F analysis? Generally, some of the p values are between 0.6 and 0.1 suggesting a failure to power the study sufficiently. What are the effect sizes? How did the authors determine the sample size a priori?

Yes, the excluded participants include both females.

We did not determine sample size *a priori* due to the inherent complexity of our design, which depends on both the number of participants and the number of motor units identified per participant. However, the number of identifiable motor units is not easy to predict, making it challenging to determine sample size *a priori*, particularly when using linear mixed-effects model. Instead, we based our sample size on a comparable previous study (Martinez-Valdes et al., 2020 [n=15]) that used surface EMG decomposition and successfully detected significant changes in motor unit discharge rate in the presence of pain. Given our use of four grids of electrodes, we anticipated identifying at least as many motor units as reported by Martinez-Valdes et al. (2020). Indeed, we identified an average of 28.6 ± 14.8 motor units (range: 7-53) motor units per condition and participant during the trapezoidal contractions, resulting in a total of 288 motor units successfully tracked across conditions. This exceeds typical yields reported for the vastus lateralis muscle in previous studies.

While our main outcomes are adequately powered, we recognize that, as is common in experimental studies, non-significant findings—particularly those with p-values between 0.05 and 0.1 (geometric measures of neuromodulation)—may reflect insufficient power to detect smaller effects. This is discussed as follows: “[...] we used complementary metrics that assess the deviations from a linear increase in discharge rate during the ramp-up phase of the triangular contractions (acceleration, brace height, and attenuation) (Beauchamp et al., 2023). None of these metrics were significantly modified during Pain. **While it does not definitely rule out an effect of pain on neuromodulation, it is noteworthy that these metrics were derived from a large sample of motor units (n=145), suggesting that neuromodulation is unlikely a major factor in the observed changes in motor unit discharge rate during transient experimental pain. However, this requires confirmation in a larger sample size**”

References:

Martinez-Valdes E, Negro F, Farina D, Falla D. Divergent response of low- versus high-threshold motor units to experimental muscle pain. *J Physiol.* 2020 ; 598(11): 2093-2108.

Pg 20: In Figure 4A, the proportion of common input is about 0.3. In the simulated data, Figure 5F, the proportion of common input is around 0.7 or above, even in the clustered and heterogenous high variance conditions. You state that the simulation needs to match all four experimental results, but I don't think any of the models reproduced the PCI data. So how well is the model mimicking the physiology? This should be discussed.

We appreciate the opportunity to discuss this point further. It is important to note that computational models - not just those of motor unit - simplify complex physiological systems and cannot perfectly replicate biological reality due to our incomplete understanding of all underlying physiological mechanisms. Our model of motor units aimed to test specific hypotheses rather than perfectly replicate every aspect of motor unit behaviour. Despite incorporating numerous physiological parameters (membrane capacitance, conductance, inhibitory and excitatory currents, etc; see section 2.6) that contribute to motor neuron behaviour, accurately capturing the complex interactions between these parameters is impossible due to incomplete understanding of the underlying neurophysiological processes at the origin of the behaviour of motor unit populations. In addition, the strength of computational models lies primarily in their capacity to capture qualitative behavior and provide a conceptual framework for understanding the phenomenon under study. Although

more complex models may provide a closer quantitative fit to experimental data, they often at the cost of introducing numerous additional parameters. This increase in complexity expands the model's degrees of freedom exponentially, potentially sacrificing parsimony and interpretability, and thus leading to more opaque, 'black-box' explanations.

Our approach aimed to replicate the key experimental outcomes (listed in Lines 752-755), including realistic discharge rates, decrease in discharge rate due to inhibitory inputs, and changes in variance explained by PC1 and the proportion of common input. To ensure full transparency and reproducibility, the simulation code and the inputs parameters have been made publicly available at:

https://github.com/FrancoisDeroncourt/Pain_inhibition_simulation

The following text has been added in the limitation section of the Discussion: "***Fourth, we acknowledge that our simulation model provides a simplified representation of the complex biophysical processes underlying motor unit behaviour. This simplification likely accounts for certain discrepancies between simulation and experimental results (e.g. higher values of proportion of common inputs for the simulation results, Fig. 5E). Nevertheless, the model successfully reproduced the key experimental outcomes, including realistic discharge rates, the decrease in discharge rate due to inhibitory inputs, and the change in proportion of common input.***"

Pg 20: Figure 5D: What is the x axis on this Figure?

There is no defined x-axis in Figure 5D. Data points are horizontally arranged to prevent overlap and to illustrate their distribution.

Pg 24, last para: "by multiple latent factors defining a multidimensional manifold." Could this sentence be written with less jargon initially and then expanded upon with the technical language? This would improve the understanding for the reader without expertise in that technique.

The text has been revised as follows: "*This concept is particularly well illustrated in the vastus lateralis muscle (the muscle studied here), where the activity of motor units is represented by **multiple underlying patterns (latent factors)** (Del Vecchio et al., 2023; Deroncourt et al., 2024).*"

Referee #2:

The present study aimed to examine the distribution pattern of inhibitory inputs to the vastus lateralis motor unit pool in response to pain during low/moderate knee-extension contraction. The article is well-written, and the experiments were conducted with rigor, incorporating multiple insightful analyses. The findings provide novel evidence of the heterogeneous distribution of nociceptive inputs to a pool of lower-threshold motor units. While these results are compelling, there are some areas that require the authors' attention.

My primary concern with this study is the force level assessed, and the strong conclusions drawn despite only evaluating a single force level. While previous studies demonstrating reductions in discharge rate with pain have been conducted at similar force levels (10%-30% MVC), assumptions regarding differential inhibition across the motor unit pool (i.e., lower versus higher threshold motor units) have been based on much higher force levels (~70% MVC). In the present study, the authors examined the quadriceps, a muscle with a high upper recruitment limit (approaching 100% MVC). As a result, the assumptions made here primarily apply to lower-threshold motor units, rather than the full motor unit pool. This distinction should be clearly reflected in the title and throughout the manuscript. Although the results remain valuable, this study does not fully address heterogeneous inhibitory (or excitatory) inputs across the entire motor unit pool, particularly for higher-threshold motor units, which were not assessed. Additionally, several relevant studies on changes in common synaptic inputs should be incorporated into the discussion. Finally, some additional analyses (outlined below) could further strengthen the study's implications.

We thank the reviewer for their thorough review of our manuscript and their insightful comments, which guided changes that have strengthened the manuscript. These are detailed in the following.

We fully agree that our results are limited to a moderate contraction intensity only, i.e. 30% of MVC. We believe that moderate contraction intensities are representative of those typically encountered during daily activities and still allow for the assessment of the behaviour of motor units with different sizes, albeit within the lower range. Of note, non-uniform inhibition across the active motor units have been suggested from studies performed at higher force levels (~70% MVC; Martinez-Valdes et al., 2020), but also from a study performed at very low-force (Hodges et al., 2021). As requested by the reviewer, we have clearly emphasized this focus on moderate contraction intensity in the revised manuscript (e.g. abstract, graphical abstract, introduction, concluding paragraphs in the Discussion); specific changes are detailed in our responses to specific comments below. The term “moderate” was chosen to contrast with the low-contraction intensities (5-10% MVC) typically used in experimental pain studies investigating motor unit discharge characteristics.

Of note, we revised the title to avoid making a broad generalisation: “**Non-homogeneous distribution of inhibitory inputs among motor units in response to nociceptive stimulation**”. However, for the sake of clarity and conciseness, we opted not to include information about contraction intensity in the title, but we clearly stated this information in the abstract and throughout the manuscript. However, if both the reviewer and the editor believe this detail must be included, we would be happy to do so.

Reference:

Hodges PW, Butler J, Tucker K, MacDonell CW, Poortvliet P, Schabrun S, Hug F, Garland SJ. Non-uniform Effects of Nociceptive Stimulation to Motoneurons during Experimental Muscle Pain. *Neuroscience*. 2021 21; 463:45-56.

Other minor comments below:

- Please add line numbers

This has been done as requested.

- Inform force intensity assessed in the abstract

The text has been revised as follows: “*This study combined experimental data and in silico models to investigate the contribution of inhibitory and neuromodulatory inputs to motor unit behaviour in response to nociceptive stimulation during contractions at 30% of maximal torque.*”

- One important aspect that is completely overlooked in the introduction is that the effects of pain on motor unit firing properties are both force- and velocity-dependent. The authors present previous evidence as if studies report conflicting responses, but they should also acknowledge that these differences may be influenced by variations in force intensity and task demands. For example, Martinez-Valdes et al. (2020) observed reductions in discharge rate at low forces (consistent with many previous studies) but reported maintained or even increased discharge rates at higher force levels. The authors should discuss these variations across studies and consider how force intensity and task-specific factors might influence the motor unit firing properties assessed in response to pain.

Thank you for this comment. We have clarified the details about contraction intensity in the introduction, as follows:

Line 105: “*However, this overall (average) decrease in motor unit discharge rate, **observed during low- to moderate-intensity force-matched contractions**, is often accompanied by an increase in the discharge rate of a small proportion of motor units and the recruitment of new units (Tucker et al., 2009; Hug et al., 2013; Martinez-Valdes et al., 2021).*”

Line 114: “*Martinez-Valdes et al. (2020) observed a **divergent response to nociceptive stimulation between low- and high-threshold motor units during contractions performed at a relatively high intensity (70% of maximal voluntary contraction [MVC])**.*”

Line 152: *“In this study, we identified a large sample of motor units (up to 53 per participant per contraction) by decomposing high-density electromyographic (EMG) signals collected during moderate-intensity isometric contractions.”*

- Input to simulated neurons: 'about 12 N.m for the number of simulated motor units.' - Does this refer to motor unit twitch force? Or Knee extension torque? clarify.

We apologize for the lack of clarity. This refers to the torque that the active motor units were required to generate to reach 30% of maximal torque (MVC) in the simulated data. However, this information is not essential, and for clarity, the text has been revised as follows *“We estimated the common excitatory input signal required for active motor units to match a target force plateau at 30% of MVC, i.e. about 12 N.m for the number of simulated motor units”*.

Of note, for transparency and reproducibility, the simulation code along with the input parameters has been made publicly available at:

https://github.com/FrancoisDernoncourt/Pain_inhibition_simulation

- In my opinion, the hypothesis tested by Martinez-Valdes et al. 2020, cannot be tested at 30% MVC. A large proportion of the MU pool of the VL is unfortunately not assessed at this force level.

According to the reviewer's comment, we modified the text as follows: *“They proposed that this divergent effect arises from a non-homogeneous distribution of inhibitory afferent inputs, with greater inhibition directed toward low-threshold units; however, this has not yet been directly confirmed. It is important to note that these results are also compatible with a homogeneous distribution of inhibitory inputs, as such homogeneous inhibition would cause a greater inhibitory post-synaptic potential in low-threshold units (Luscher et al., 1979).”*

- Pain: What was the pain duration? Did the pain intensity vary significantly while recording the multiple contractions? Did pain intensity have an effect on the motor unit properties assessed in the present study?

We did not record pain intensity at fixed time points; instead, participants rated their pain levels before and during each contraction. These data reported in the Results section and in Fig. 1. On average pain intensity followed a pattern very similarly to that reported in Fig. 1 of Martinez-Valdes et al. (2020).

The reviewer asks a valid question regarding the relationship between pain level and motor unit behaviour. The correlation coefficient between the change in discharge rate from Control to Pain and the pain level assessed during contractions was -0.45, although this was not statically significant ($p=0.091$). This information has been added in the Results section: *“The change in discharge rate between Control and Pain was not significantly correlated with the pain level reported during contraction ($r=-0.45$; $p=0.091$)”*.

- Change to interference EMG

Changed as suggested. Thank you.

- 'The mean EMG amplitude calculated over the grid' - wasn't this calculated for all the 4 grids?

The reviewer is correct. The text has been modified as follows: *“The mean EMG amplitude calculated over the four grids did not significantly differ between conditions ($F(2,28)=0.91$, $p=0.42$).”*

- Any association between the level of heterogeneity in inputs and pain level experienced? - in other words, would the participants experiencing more pain show higher or lower heterogeneity? This would help understand the functional implication of the findings. There was no significant correlation. This information has been added in the Results section: *“Despite the significant decrease in motor unit discharge rate during Pain, a notable proportion of motor units did not exhibit such a decrease, i.e. 24.8% of motor units during trapezoidal contractions (Figure 2), with no significant correlation with the reported pain level during contraction ($r=-0.27$; $p=0.32$).”*

Discussion

- A key finding of this study is that, despite the overall reduction in firing rate in response to experimental pain, there appears to be a heterogeneous distribution of inhibitory and excitatory inputs across the motor unit pool. However, the authors should emphasize that these changes are primarily observed in lower-threshold motor units. Comparisons with Martinez-Valdes et al. (2020) are somewhat challenging, as that study also reported a similar reduction in discharge rate at 20% MVC but divergent behaviors at 70% MVC. However, unlike the present study, it did not assess the differential distribution of inputs across the pool of lower threshold motor units. Therefore, I believe the current study provides several additional insightful analyses that contribute meaningfully to this area of research, but claims need to be more specific to the population of units assessed.

We agree with the reviewer on the need for more cautious interpretation. Because motor unit adaptation to pain appears more directly influenced by contraction intensity than motor unit size (low threshold units exhibit different behaviour at 20% and 70% MVC in Martinez-Valdes et al. 2020, with very weak correlation between change in DR and recruitment threshold), we chose to emphasize the limitation related to contraction intensity.

The first paragraph of the discussion has been revised as follows: *“We investigated the behaviour of large samples of motor units during submaximal contractions at 30% of MVC under nociceptive stimulation to identify inhibitory patterns within the population of active motor units. Our experimental data, combined with in silico modelling, demonstrate an increase in inhibition during such moderate-intensity contractions, which is non-homogeneously distributed across motor units [...]”*.

This is also discussed in the limitation section: *“Second, to further avoid fatigue and because ΔF validation for estimating the magnitude of persistent inward currents is not established at high force levels, our protocol was limited to contractions at 30% of MVC. Even though our results need to be confirmed across a broader range of forces, we believe that this limitation does not impact our main conclusion that a noxious stimulus induces non-homogeneous inhibition within a group of motor units with varying recruitment thresholds.”*

The first sentence of the concluding paragraph has also been modified: *“To conclude, this study provides novel insights into the distribution of inhibitory inputs to motor units during moderate-intensity contractions performed in the presence of experimental pain.”*

Finally, to acknowledge that we did not examine the full pool of motor units, we have, in several instances throughout the manuscript, replaced “motor unit pool” with “population of active motor units” or simply by “across motor units”.

- Regarding common input, it is surprising that the studies by Farina et al. (J Neurophysiol, 2011) and Yavuz et al. (J Neurophysiol, 2015) are not included among the references. These studies reported findings that contrast with those presented here and should be discussed in relation to the current results. In particular, PCI primarily reflects changes in low-frequency coherence (<5 Hz), whereas Farina et al. (2011) found an increase in this frequency range in response to pain. It is also unexpected that, despite no observed variations in torque steadiness, the authors report a significant decline in PCI and a reduction in the variance explained by PC1. What mechanisms might underlie these differences? Additionally, do the authors observe changes in CST versus torque cross-correlation? Examining this relationship could provide insight into potential compensatory strategies employed during task execution.

With all due respect, we do not believe that the studies mentioned by the reviewer demonstrate an increase in the level of common input. It is important to differentiate between the power spectral density of the CST and the coherence (or correlation) among motor units. Specifically, the power of lower-frequency components may change independently from the coherence across motor units at those lower-frequencies, and thus from the level of common inputs. This is illustrated by the findings of Yavuz et al. (2015, cited by the reviewer), who reported an overall increase in power of the neural drive accompanied by a significant decrease in coherence in the full bandwidth (short-term synchronisation). Of note, the study by Farina et al. (2012) did not report changes in motor unit short-term synchronisation with

pain. However, as this metric reflects the full frequency bandwidth, it may have masked specific alterations within the lower-frequency band. Overall, we believe that the observed decrease in proportion of common input and variance explained by PC1 in our study do not contradict previous studies.

We thank the reviewer for bringing our attention to these articles. The revised version includes the citation of the work from Yavuz et al. (2015): "*The **reduced** proportion of common input reported in our experimental data is **consistent with previous findings reporting a decrease in motor unit short-term synchronisation during pain (Yavuz et al., 2015)***".

As requested by the reviewer, we calculated the correlation between the cumulative spike train and effective force, which did not significantly differ across conditions. The following text has been added:

In the Methods section: "*Finally, the cumulative spike train was computed by summing the spike trains of all active motor units. It was then low-pass filtered at 2.5 Hz and correlated with the effective force.*"

In the Results section: "*Of note, the correlation between effective force and the cumulative spike train did not differ significantly across conditions ($p=0.865$; 0.59 ± 0.16 for Control, 0.59 ± 0.16 for Pain, and 0.59 ± 0.16 for Washout).*"

- 'Martinez-Valdes et al. (2020) observed a divergent response between low- and high-threshold units' - Emphasise that this was only found at high forces, at low forces (20% MVC) Martinez-Valdes et al., 2020 found the same reductions in discharge rate as the ones reported in the present study.

To address the reviewer's comment, the text has been modified as follows: "*For instance, Martinez-Valdes et al. (2020) observed a divergent response between low- and high-threshold motor units **during high-intensity contractions (70% of MVC), with high-threshold units increasing their discharge rate during pain, while the discharge rate of lower-threshold remained unchanged.***"

- The assumptions made by Mesquita et al. 2020, also consider higher forces, therefore, the PIC findings of the present study do not fully explain the potential differential inhibition/excitation among lower and higher threshold units.

If the reviewer refers to the sentence on Line 742-744, the citation of Mesquita et al. was incorrect and has been removed in the revised version.

- 'These simulation findings strongly support a non-homogeneous distribution of inhibitory inputs to motor units, independent of their size' - Yes, but only at low to moderate force levels. What happens at higher forces remains unknown.

Thank you. The text has been modified as follows: "*These simulation findings strongly support that, **during moderate-intensity contractions, inhibitory inputs are non-homogeneously distributed among motor units, independent of their size***".

- "This may explain why newly recruited higher-threshold units during pain do not always follow the expected orderly recruitment pattern (Tucker et al., 2009)." Were the authors able to identify newly recruited motor units in response to pain? To date, Tucker et al. (2009) remain the only study to report such changes, but their findings were based on more limited methodologies. Given the larger motor unit samples obtained in the present study, it would be valuable to assess whether the number of identified units differs across conditions. Specifically, were unique motor units recruited in the pain condition? Did you see a violation of the size principle? Analyzing this aspect could provide further insights into the recruitment strategies underlying motor unit behavior in response to pain.

We appreciate the opportunity to discuss this point further.

First, we agree with the reviewer that identifying the derecruitment of motor units and the recruitment of additional motor units during the Pain condition would have been valuable. However, as mentioned in our Limitations section, the surface EMG decomposition approach does not allow for conclusive interpretations regarding motor unit recruitment and derecruitment. This is because the inability to track a motor unit across conditions should not be directly interpreted as a change in recruitment, as it may result from the decomposition

algorithm failing to identify the same motor unit between conditions. In other words, while tracking methods based on MUAP shapes (Martinez-Valdes et al., 2017) or the re-application of motor unit filters (used in our study) can reliably track the same units across conditions, they cannot confidently identify newly recruited or derecruited motor units.

Second, the reviewer raised an interesting point regarding potential violations of the size principle, which should indeed be observed in the case of non-uniform inhibition. We initially explored this question by calculating the Spearman's rank correlation to compare motor unit recruitment order across conditions. Because this analysis had to be conducted at the individual participant level, we excluded some participants who did not have a sufficient number of motor units (datapoints) for a meaningful correlation analysis. In the remaining nine participants, we observed a trend toward lower correlations between Control and Pain than between Control and Washout ($p=0.065$), suggesting that deviations from the size principle may have occurred more frequently during Pain. However, given the small sample size, we remain cautious and do not consider this result conclusive at this stage. We are currently conducting a follow-up study designed to test this hypothesis more robustly.

References:

- Martinez-Valdes E, Negro F, Laine CM, Falla D, Mayer F, Farina D. Tracking motor units longitudinally across experimental sessions with high-density surface electromyography. *J Physiol*. 2017 Mar 1;595(5):1479-1496.
- Avrillon S, Hug F, Enoka RM, Caillet AHD, Farina D. The identification of extensive samples of motor units in human muscles reveals diverse effects of neuromodulatory inputs on the rate coding. *Elife*. 2024;13:RP97085

Dear Dr Hug,

Re: JP-RP-2025-288504R1 "Non-Homogeneous Distribution of Inhibitory Inputs Among Motor Units in Response to Nociceptive Stimulation" by Francois Hug, Francois Deroncourt, Simon Avrillon, Jacob Thorstensen, Manuela Besomi, Wolbert van den Hoorn, and Kylie J Tucker

Thank you for submitting your manuscript to The Journal of Physiology. It has been assessed by a Reviewing Editor and by 2 expert referees and we are pleased to tell you that it is acceptable for publication following satisfactory revision.

REVISION CHECKLIST:

We look forward to receiving your revised submission.

Yours sincerely,

Richard Carson
Senior Editor
The Journal of Physiology

EDITOR COMMENTS

Reviewing Editor:

The authors have comprehensively addressed the Reviewers' comments. Reviewer 2 has raised two relatively minor points for further consideration and clarification before publication.

REFEREE COMMENTS

Referee #1:

The authors have addressed my original comments. I have no further comments.

Referee #2:

Dear authors, thank you for the detailed revision of your study. The additions made have improved consistency, however, a few minor issues remain.

Title: I believe it's important to adjust the title to accurately reflect what was actually measured, in order to avoid potential confusion-especially considering that some of the motor unit behaviors reported may not be representative of the entire motor unit pool. Additionally, your study is considerably stronger and methodologically distinct from that of Hodges et al., which drew conclusions about the non-homogeneous effects of pain on motor units based on data from different participants. For this reason, it's essential to be precise about what was measured and the specific effects observed. I believe that the lack of measurement of a higher force level is a major limitation of the current study and I'm surprised this was not done as this would have strengthened the author's conclusions. Therefore, I believe the title should be revised to more accurately reflect the specific changes observed within the motor unit pool assessed in this study.

Common input:

Considering the authors' response regarding common input: have you computed the PCI (Negro et al., 2016) across the full

bandwidth (i.e., 0-50 Hz)? PCI was originally developed to quantify the proportion of common input at low frequencies (0-5 Hz). If I understood correctly, lines 345-347 suggest that you did indeed calculate PCI within this low-frequency range: "We modelled the relationship between the average coherence values in the 0-5 Hz bandwidth and the number of motor units in each group using a least-squares curve fitting approach, based on the two-parameter model described by Negro et al. (2016b)."

If so, this measure is very similar to quantifying motor unit coherence in the delta band. As noted in my previous review, delta-band coherence was shown to increase in response to pain in the study by Farina et al., likely driven by the increase in force fluctuations observed under painful conditions-an effect not present in your study (i.e., no significant change in force output).

In contrast, short-term synchronization is typically associated with coherence changes in the beta band (Moritz et al., J Neurophysiol, 2005). However, neither Farina et al. nor Yavuz et al. reported changes in beta-band coherence with pain. It's also important to clarify that the short-term synchrony estimation used by Yavuz et al. is based on the relationship between the time-domain peak in the cross-correlation and the frequency-domain integral across the full spectrum. Still, this is not a direct time-domain measurement of short-term synchrony, as commonly defined (see Sears and Stagg, 1976). Given that your analyses focus primarily on low-frequency components, it would not be appropriate to attribute the observed changes to alterations in short-term synchrony.

My previous question was aimed at understanding why PCI changed in your study despite the absence of changes in force output. In contrast, the findings reported by Farina et al. and Yavuz et al. are likely driven by pain-induced increases in force variability. This makes your results particularly intriguing. To be clear, I am not questioning the validity of your findings-it's well recognized that changes in force variability in response to experimental pain are not consistently observed across studies (Arvanitidis et al. Eur J Pain 2025). Nevertheless, the observation that PCI decreased without any accompanying variation in force output raises interesting questions. Do the authors have any hypotheses or potential mechanisms that could explain this dissociation?

'($\rho=0.865$; 0.59 ± 0.16 for Control, 0.59 ± 0.16 for Pain, and 0.59 ± 0.16 for Washout).' - All these correlations seem to be exactly the same. Are these values correct?

END OF COMMENTS

REFEREE COMMENTS

Referee #1:

The authors have addressed my original comments. I have no further comments.

Referee #2:

Dear authors, thank you for the detailed revision of your study. The additions made have improved consistency, however, a few minor issues remain.

Title: I believe it's important to adjust the title to accurately reflect what was actually measured, in order to avoid potential confusion-especially considering that some of the motor unit behaviors reported may not be representative of the entire motor unit pool. Additionally, your study is considerably stronger and methodologically distinct from that of Hodges et al., which drew conclusions about the non-homogeneous effects of pain on motor units based on data from different participants. For this reason, it's essential to be precise about what was measured and the specific effects observed. I believe that the lack of measurement of a higher force level is a major limitation of the current study and I'm surprised this was not done as this would have strengthened the author's conclusions. Therefore, I believe the title should be revised to more accurately reflect the specific changes observed within the motor unit pool assessed in this study.

The title has been changed to: « Non-Homogeneous Distribution of Inhibitory Inputs Among Motor Units in Response to Nociceptive Stimulation at Moderate Contraction Intensity ».

Common input:

Considering the authors' response regarding common input: have you computed the PCI (Negro et al., 2016) across the full bandwidth (i.e., 0-50 Hz)? PCI was originally developed to quantify the proportion of common input at low frequencies (0-5 Hz). If I understood correctly, lines 345-347 suggest that you did indeed calculate PCI within this low-frequency range: "We modelled the relationship between the average coherence values in the 0-5 Hz bandwidth and the number of motor units in each group using a least-squares curve fitting approach, based on the two-parameter model described by Negro et al. (2016b)."

We confirm that PCI was calculated within the low-frequency range.

If so, this measure is very similar to quantifying motor unit coherence in the delta band. As noted in my previous review, delta-band coherence was shown to increase in response to pain in the study by Farina et al., likely driven by the increase in force fluctuations observed under painful conditions-an effect not present in your study (i.e., no significant change in force output).

With all our due respect, we do not believe that Farina et al (2011 - PMC3289482) reported an increase in delta-band coherence. They reported no difference between conditions for either the common drive index or the common input strength (even though slightly different from coherence within the low-frequency range). To our understanding, they did not compute coherence in their analyses.

In contrast, short-term synchronization is typically associated with coherence changes in the beta band (Moritz et al., J Neurophysiol, 2005). However, neither Farina et al. nor Yavuz et al. reported changes in beta-band coherence with pain. It's also important to clarify that the short-term synchrony estimation used by Yavuz et al. is based on the relationship between the time-

domain peak in the cross-correlation and the frequency-domain integral across the full spectrum. Still, this is not a direct time-domain measurement of short-term synchrony, as commonly defined (see Sears and Stagg, 1976). Given that your analyses focus primarily on low-frequency components, it would not be appropriate to attribute the observed changes to alterations in short-term synchrony.

The reviewer is right that short-term synchronization is not directly related to the level of common inputs at lower frequencies. The following sentence has been removed from the Discussion section: *“The reduced proportion of common input reported in our experimental data is consistent with previous findings reporting a decrease in motor unit short term synchronisation during of pain (Yavuz et al., 2015).”*

My previous question was aimed at understanding why PCI changed in your study despite the absence of changes in force output. In contrast, the findings reported by Farina et al. and Yavuz et al. are likely driven by pain-induced increases in force variability. This makes your results particularly intriguing. To be clear, I am not questioning the validity of your findings-it's well recognized that changes in force variability in response to experimental pain are not consistently observed across studies (Arvanitidis et al. Eur J Pain 2025). Nevertheless, the observation that PCI decreased without any accompanying variation in force output raises interesting questions. Do the authors have any hypotheses or potential mechanisms that could explain this dissociation?

We appreciate the opportunity to discuss this point further. As mentioned in our response in the previous round of review, we do not believe that the fact that PCI decreased, despite the absence of change in force output, contradicts results from Farina and Yavuz. To the best of our knowledge, there is no strong relationship between the proportion of common input and force variability as illustrated in the figure below, where two different proportions of common input result in the same force variability (Farina and Negro, 2015 – ESSR).

Figure 7. Motor unit synchronization and force variability. A pool of 120 simulated motor neurons receive a common and an independent input of power ratio 50% (upper traces) and 25% (lower traces). As expected, the synchronization estimated by cross histogram of pairs of motor units is greater (common input strength (CIS) = 1.52) in the first than in the second case (CIS = 0.70), as also shown by representative cross histograms. Despite the different ratio between common and independent input and different CIS values in the two cases, the variability in the generated force by the motor neuron pool is very similar between the two conditions (CoV for force, 3.8% vs 3.7%). AU indicates arbitrary units.

'(p=0.865; 0.59{plus minus}0.16 for Control, 0.59{plus minus}0.16 for Pain, and 0.59{plus minus}0.16 for Washout).' - All these correlations seem to be exactly the same. Are these values correct?

Thank you for highlighting this mistake. The values have been corrected, and the text now reads: *“Of note, the correlation between effective force and the smoothed cumulative spike train did not differ significantly across conditions (p=0.865; 0.59±0.16 for Control, 0.58±0.13 for Pain, and 0.60±0.14 for Washout).”*

Dear Professor Hug,

Re: JP-RP-2025-288504R2 "Non-Homogeneous Distribution of Inhibitory Inputs Among Motor Units in Response to Nociceptive Stimulation at Moderate Contraction Intensity" by Francois Hug, Francois Deroncourt, Simon Avrillon, Jacob Thorstensen, Manuela Besomi, Wolbert van den Hoorn, and Kylie J Tucker

We are pleased to tell you that your paper has been accepted for publication in The Journal of Physiology.

Yours sincerely,

Richard Carson
Senior Editor
The Journal of Physiology

If you would like to receive our 'Research Roundup', a monthly newsletter highlighting the cutting-edge research published in The Physiological Society's family of journals (The Journal of Physiology, Experimental Physiology, Physiological Reports, The Journal of Nutritional Physiology and The Journal of Precision Medicine: Health and Disease), please click this link, fill in your name and email address and select 'Research Roundup':
<https://www.physoc.org/journals-and-media/membernews>

- **TRANSPARENT PEER REVIEW POLICY:** To improve the transparency of its peer review process, The Journal of Physiology publishes online as supporting information the peer review history of all articles accepted for publication. Readers will have access to decision letters, including Editors' comments and referee reports, for each version of the manuscript as well as any author responses to peer review comments. Referees can decide whether or not they wish to be named on the peer review history document.
- You can help your research get the attention it deserves! Check out Wiley's free Promotion Guide for best-practice recommendations for promoting your work at: www.wileyauthors.com/eoo/guide. You can learn more about Wiley Editing Services which offers professional video, design, and writing services to create shareable video abstracts, infographics, conference posters, lay summaries, and research news stories for your research at: www.wileyauthors.com/eoo/promotion.
- **IMPORTANT NOTICE ABOUT OPEN ACCESS:** To assist authors whose funding agencies mandate public access to published research findings sooner than 12 months after publication, The Journal of Physiology allows authors to pay an Open Access (OA) fee to have their papers made freely available immediately on publication.

EDITOR COMMENTS

No further comments.

REFEREE COMMENTS

Referee #2:

I thank the authors for their thorough revision and insightful discussions. I have no more comments to make.